

# An in situ observation dataset of soil hydraulic properties and soil moisture in a high and cold mountainous area on the northeastern Qinghai-Tibet Plateau

Jie Tian[1], Baoqing Zhang[1*], Xuejin Wang[1], Chansheng He[1,2*]

[1]Key Laboratory of Western China's Environmental Systems (Ministry of Education), College of Earth and Environmental Sciences, Lanzhou University, Lanzhou, Gansu 730000, China

[2]Department of Geography, Western Michigan University, Kalamazoo, MI 49008, USA

*Correspondence to*: Chansheng He (he@wmich.edu) and Baoqing Zhang (baoqzhang@lzu.edu.cn)

**Abstract.** Soil hydraulic properties (SHPs) and soil moisture (SM) are fundamental to describing and predicting water and energy cycles at the land surface, and for regulating evapotranspiration, infiltration and runoff. However, information about these soil properties from existing datasets is often scarce and inaccurate for high and cold mountainous areas such as the Qinghai-Tibet Plateau (QTP), which hampers our understanding of hydrological and energy cycle processes over large mountainous areas like the QTP. Based on soil profile data at depths of 5 cm and 25 cm from 238 sampling sites, and on soil data from 32 SM monitoring stations at depths of 5 cm, 15 cm, 25 cm, 40 cm, and 60 cm, we have compiled a SHP and SM dataset for a high and cold mountainous area, Northeastern QTP. We used this dataset to explore the large-scale spatial and temporal variability of SHPs and of SM across the study area. Our evaluation of several existing SHP datasets, SM datasets derived from remote-sensing, reanalysis and data assimilation, showed that SHPs (soil texture, bulk density, and soil saturated hydraulic conductivity) in these datasets are biased, and do not capture the spatial variability recorded in the in-situ observations. When comparing with the in-situ SM observations, the SM product derived from remote-sensing was more reliable than the SM product derived from reanalysis data (which had a higher bias), and than the data assimilation product (which did not capture SM temporal variability). The in situ observation dataset presented here provides unique and important information about the SHP variability and long-term SM trends at a large-scale, high and cold mountainous area, and thus offers opportunity for further understanding of water cycle and energy exchange processes over the QTP.



## 1. Introduction

Soil provides an important link between atmospheric and hydrologic processes, and has a strong influence on surface energy and water fluxes, and on regional climate (Amundson et al., 2015; Jia et al., 2020). Soil hydraulic properties (SHP) control

water movement in the soil (Maxwell and Condon, 2016; Vereecken et al., 2015), and impact water and energy exchange between the land and the atmosphere (McColl et al., 2017; Wang et al., 2020). Soil moisture (SM) is an essential climate variable (Global Climate Observing System, 2010) because of the effects it has on the weather (Tuttle and Salvucci, 2016), on runoff (McDonnell et al., 2018), on evapotranspiration (Jung et al., 2010), on groundwater recharge (McColl et al., 2017), on the carbon cycle (Green et al., 2019; Wei et al., 2021), and on temperature extremes (Seneviratne et al., 2010). Soil properties

are represented in models at different scales, from small-scale hydrological models to regional- and global-scale Earth system models (Bai et al., 2020; Vereecken et al., 2016), and much effort has gone into assessing and improving the accuracy of model estimates of SHP and SM (e.g., Blöschl et al., 2019; Dorigo et al., 2021; Montzka et al., 2017) over the past decades. However, the uncertainty associated with information about SHP and SM over large spatial scales remains high (Benninga et al., 2018; Fatichi et al., 2020), especially for high and cold mountainous areas, such as the Qinghai-Tibet Plateau (QTP, Che et al., 2019;

Jin et al., 2015; Li et al., 2018a; Liu et al., 2021b; Zhao et al., 2018).

Due to its harsh environment, large-scale soil-sampling is rarely undertaken in high and cold mountainous areas such as the QTP (Che et al., 2019; Li et al., 2017; Li et al., 2018b; Zhang et al., 2021). This makes the data in many widely-used soil databases for mountainous areas highly uncertain (Dai et al., 2019; Wang et al., 2021). Recently, there has been considerable progress towards creating large-scale SHP datasets for the mountainous areas of the QTP. These new datasets include the soil

organic carbon (Ding et al., 2019; Song et al., 2016; Wang et al., 2021; Yang et al., 2016), soil thickness (Yang et al., 2016; Zhang et al., 2016), and soil texture (Li et al., 2018c; Lu et al., 2017), which have improved our understanding of the spatial distribution and estimation accuracy of SHP datasets over the QTP (Li et al., 2020). Field sampling and laboratory measurements are more difficult and time consuming for some key SHPs, such as the soil water retention curve and saturated hydraulic conductivity, than the equivalent investigations for more basic soil parameters, such as texture (He et al., 2021; Tian

et al., 2017). To date, very few large-scale studies have provided comprehensive in-situ measurements of both these key soil hydraulic properties and basic soil information, and there are few in situ observations of large-scale soil hydraulic properties available for the QTP (Liu et al., 2021; Tian et al., 2017; Zhao et al., 2018). The relationships between individual SHPs and their large-scale spatial distribution over the QTP remain largely unknown (Tian et al., 2017; Zhao et al., 2018). Large-scale estimates of key SHPs over the QTP, which are always made using pedotransfer function, have been shown to have particularly

high uncertainty (Dai et al., 2019; Lu et al., 2020; Van Looy et al., 2017; Zhang et al., 2018a; Zhao et al., 2018).

SM can be quantified using in-situ instruments (Dorigo et al., 2021; Vereecken et al., 2008), remote sensing (Mohanty et al., 2017) and land-process models (Muñoz-Sabater et al., 2021). Land-process models have large biases and predictions often vary between different models (Lu et al., 2020; Xia et al., 2014; Xing et al., 2021; Zhang et al., 2021). Remote-sensing observations can be challenging over mountainous regions and SM can generally only be retrieved for the uppermost 5 cm of



the soil. (Xing et al., 2021; Zhang et al., 2019). Ground-based SM measurements are the most accurate and important means
      for developing, validating, and extrapolating spatially-contiguous data derived from satellites or from land-process models
      (Dorigo et al., 2021; Benninga et al., 2018; Xia et al., 2014). However, measuring SM in-situ over large areas is difficult,
      particularly in mountainous regions, and thus such measurements are scarce (Ochsner et al., 2013; Dorigo et al., 2021).In
      recent years, numerous long-term SM monitoring networks have been established (e.g. Dorigo et al., 2021; Ochsner et al.,
2013; Bogena et al., 2018; Bogena et al., 2021), but only a few of these are in high and cold mountain areas (Che et al., 2021;
      Su et al., 2011; Jin et al., 2014; Pellet and Hauck, 2017; Zhang et al., 2021), and this limits the improvements to remote sensing
      products and land surface models for mountainous areas (Li et al., 2021; Xia et al., 2014; Su et al., 2013).

      This study presents field-sampled SHP profiles, collected through a SM monitoring network that was established in 2013,
      and uses these to compile a SHP dataset, and a long-term SM dataset for the Qilian Mountains, on the northeastern edge of the
QTP. We used these datasets to investigate SHP and SM characteristics. We discuss the characteristics of the SHPs and the
      spatial-temporal characteristics of SM over the Qilian Mountains, and evaluate the uncertainties associated with five widely-
      used soil-property datasets and three global SM products over the study area. In section 2, we describe the field campaign and
      laboratory experiments that were used to compile the new SHPs and SM datasets, and those used for the existing datasets. We
      present our results and discussion of potential applications in sections 3 and 4, and outline our conclusions in section 5. The in
situ datasets presented here fill some geographical gaps in the current global soil databases, and will be useful to both the
      hydro-climatology research and the land-surface modeling communities, for climatic and land surface studies over the QTP.

## 2. Materials and methods

### 2.1 Study sites

      The upper stream of the Heihe River Basin is in the Qilian Mountains at the northeastern border of the Qinghai-Tibet Plateau.
It has an area of $2.7 \times 10^4$ km$^2$ and an elevation range of about 2000–5600 m (Figure 1, Li et al., 2019). The area has an annual
      rainfall that varies from 200 to 700 mm (Luo et al., 2016), annual potential evapotranspiration that ranges from 700 to 2000
      mm, and an annual mean temperature range of -3.1 °C to 3.6 °C, based on data from 1960 to 2012 (He et al., 2018). Big
      differences in climate condition and topography lead to high spatial heterogeneity in land-cover and soil properties in
      mountainous areas (Jin et al., 2015). The land-cover is mainly grassland, forestland and sparsely vegetated land (Zhou et al.,
2016). The main soil types are Calcic Chernozems, Kastanozems, and Gelic Regosols. The main soil texture classes are silt
      loam, silt and sandy loam (Tian et al., 2017; 2019). The study area has a harsh natural environment and are challenges for
      fieldwork in the high and cold mountainous areas (Figure S1).

**Figure 1. (a) Study area and its location in Northwest China, (b) distribution of the soil sample locations in the upper stream of the Heihe River Basin, on a basemap that shows the distribution of the 32 main LULC (Land Use/Cover)-soil-DEM types. (c)-(f) photographs from field sampling in the study area.**

## 2.2. Field sampling

Soil type, land-cover and elevation are all spatially heterogeneous in the Qilian Mountains (Li et al., 2019). To capture the spatial distribution of the SHPs, we obtained existing land cover/land use data (LULC, landuse/landcover data for the Heihe River Basin (2011), http://data.tpdc.ac.cn/en/), soil type data (Soil, Gansu Soil Handbook at 1:1 000 000 scale) and digital elevation model data (DEM, https://earthexplorer.usgs.gov/) for this area. We divided the study area into 32 homogeneous zones by converting, overlaying and aggregating the LULC-Soil-DEM datasets in ArcGIS (Figure 1 (b)). The procedure is described in detail in Jin et al. (2015). Thus, we divided the study area into 32 LULC-Soil-DEM zones, each with a unique set of land use/land cover, soil and elevation characteristics. These zones represent the landscape features of the mountainous study area. To further analyze the spatial distribution of the SHPs in the mountainous area, we used equation (1) to determine the minimum number of random soil samples that were needed for a robust statistical analysis (Jensen, 2005).



$$N = \frac{B \prod_i (1 - \prod_i)}{b_i^2} \tag{1}$$

Where $\prod_i$ is the percentage of classes (the $i$ class) with percentage closest to 50% among all the types (the total number of the class is $k$); B represents percentiles around $(\frac{\alpha}{k})$ of a Chi-square distribution with 1 degree of freedom; $b_i$ is the expected

precision of class $i$; and α is the confidence level. In the study area, class $i$ is the class that closest to 50% of all the types, which is saturated frigid frozen soil in dense grassland, with elevation between 3500 m and 4000 m in our stud area. We calculated N to be 339 or 170 for a precision, $b_i$, of 0.05 and a confidence level, α, of 0.95 or 0.85, respectively. The details of this procedure are presented in Li et al. (2018).

We established a long-term in-situ monitoring network to cover the mountainous area, based on the spatial distribution of

the 32 LULC-Soil-DEM zones. One long-term SM monitoring station was established in each of the LULC-Soil-DEM zones, the specific position of each station was determined by the overall distribution of the stations and road accessibility (Figure S2-Figure S3). According to soil depth survey, the soil profile was investigated from the surface to 70 cm depth, and was divided into 5 layers (0–10 cm, 10–20 cm, 20–30 cm, 30–50 cm, and 50–70 cm) at each station. SM sensors (ECH2O 5TE, METER Group Inc., USA) were installed in the center of each layer to continuously measure the SM at different soil depths

with a time interval of 30 min. These measurements began in 2013. Soil-specific sensor calibrations were performed with the direct calibration method using soil samples from each station (Cobos and Chambers, 2010; Zhang et al., 2017). A detailed description of the SM monitoring is presented in Tian et al. (2019; 2020) and Zhang et al. (2017). Since the soil freezes in winter, SM data are only available for the growing seasons (May to October, Tian et al., 2019), and we averaged the measurements to obtain monthly SM data for use in this study. Thus, monthly SM observations at different depths (5 cm, 15

cm, 25 cm, 40 cm, and 60 cm) over the growing seasons of 2014–2020 are used in this study.

To analyze the SHPs, both disturbed and undisturbed soil samples were collected from each soil layer at each station using self-sealing bags and a metal cylinder (with a diameter of 5 cm and a height of 5 cm). Environmental factors such as the position, slope, aspect, root depth, and land cover were measured at each station. Details of the installation and field measurements are shown in Tian et al. (2019). The 32 long-term SM monitoring stations constitute a large-scale in-situ SM

monitoring network in a high and cold mountainous area (Figure 1).

The calculated minimum required number of soil samples (N in Equation (1)) were collected randomly from each zone in the study area. Based on the spatial distribution of the LULC-Soil-DEM zones, road accessibility and the distribution of the soil sample locations, we collected 206 random soil sample sites in the study area from July 2012 to September 2014 (Figure 1). For each of the random sampling site, soil samples were collected at depths of 5 cm and 25 cm. At the long-term SM

monitoring stations, both disturbed and undisturbed soil samples were collected from each layer of each random site using self-sealing bags and a metal cylinder (with a diameter and height of 5 cm).

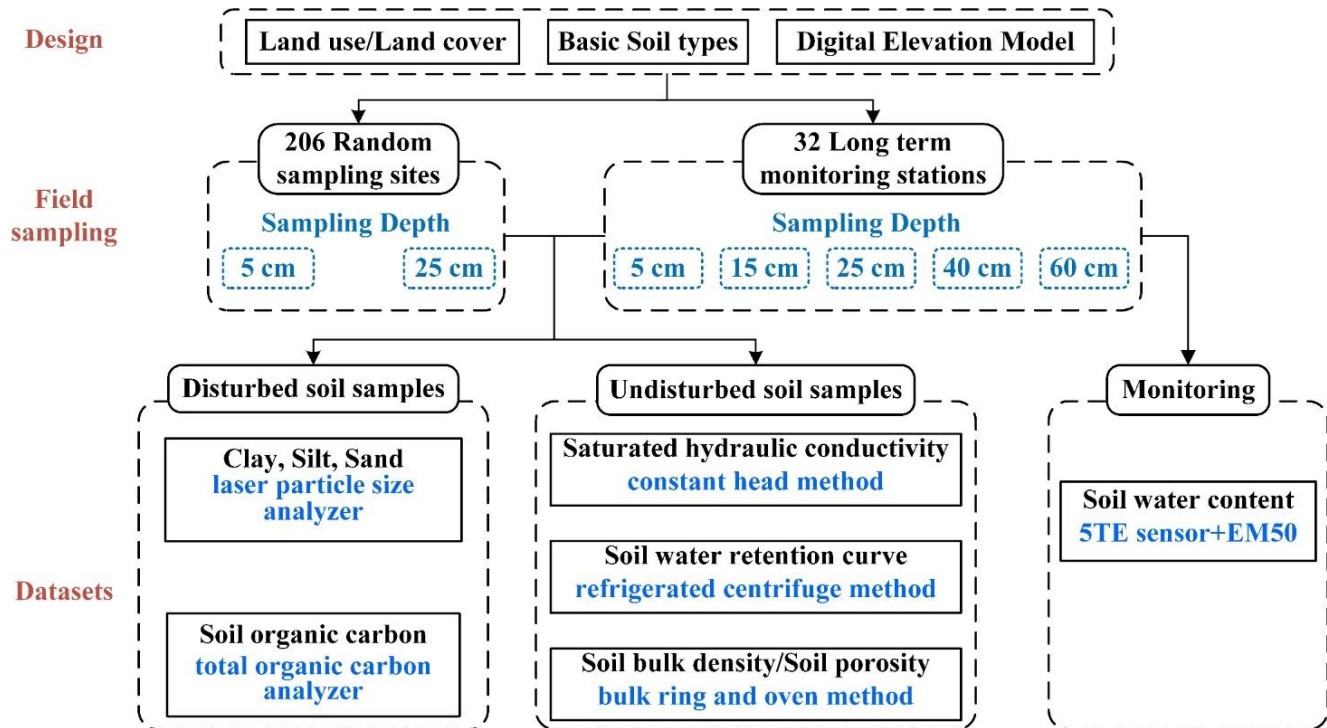

**Figure 2. The design and field sampling strategies for the datasets created in this study (the blue text describes the measurement method used for each dataset).**

### 2.3. Laboratory experiment of soil properties

The soil samples were taken to the Key Laboratory of Western China's Environmental Systems (Ministry of Education) at Lanzhou University to measure SHPs. The soil particle composition and soil organic carbon (SOC) were measured from the disturbed soil samples using a Mastersizer 2000 laser diffraction particle size analyzer (Malvern Panalytical Ltd.) and a total organic carbon analyzer (Analytik Jena GmbH), respectively. The saturated hydraulic conductivity ($K_S$) was measured for the undisturbed soil samples using the constant head method. We measured the soil water retention curve (SWRC) for the undisturbed samples using the refrigerated centrifuge method (CR-GIII High-Speed Refrigerated Centrifuge, Hitachi, Ltd.), and measured the soil bulk density for the undisturbed soil samples using the oven-drying method. Finally, soil porosity was calculated from the soil bulk density. Details of the measurements and calculations are given in Tian et al. (2017; 2019).

The experiment soil water retention curve data (measured soil water content at the specific matrix potential from 0 to 6359 hPa (also written as cm $H_2O$), were fitted using different widely-used SWRC models, and the best-fitting SWRC model was selected. The Brook-Corey (1964) and Van Genuchten (1980) models are the most widely-used SWRC models, the Kosugi (1996) model includes a specific physical mechanism (Zhang et al., 2018a), the Clapp and Hornberger (1978) model is widely used for land surface modelling (Zhao et al., 2018), and the Gardner (1970) model has the simplest form. The equations for the different SWRC models are given in the supplement.



## 2.4. Existing datasets

### 2.4.1 Soil hydraulic properties datasets

The Harmonized World Soil Database V1.2 (HWSD) is a 30 arc-second raster database that combines existing up to date regional and national soil information from all over the globe (FAO/IIASA/ISRIC/ISS-CAS/JRC, 2012). The HWSD provides an estimate of basic SHPs for two soil layers to a depth of 1 m (0–30 cm and 30–100 cm, http://www.fao.org/soils-portal/data-hub/soil-maps-and-databases/harmonized-world-soil-database-v12/en/, accessed: 10 September 2021). The SoilGrids2.0 system (SoilGrid, https://soilgrids.org/, last access: 10 September 2021) currently provides the most detailed estimate of global soil distribution, with data for multiple soil depths (0–5 cm, 5–15 cm, 15–30 cm, 30–60 cm, 60–100 cm and 100–200 cm) at a resolution of 250 m (Poggio et al., 2021). In large-scale land-process modeling for the QTP, the HWSD and SoilGrid datasets are the most widely-used datasets for basic soil properties, such as the soil texture, soil bulk density, soil organic carbon (Dai et al., 2019a; Lu et al., 2020; Su et al., 2013; Zhao et al., 2018).

The key soil hydraulic properties (key SHPs) are the soil saturated hydraulic conductivity ($K_S$) and the soil water retention curve (SWRC). A dataset developed by Yongjiu Dai using pedotransfer functions (PTFs) with a spatial resolution of 30″ (nearly 900 m) and a vertical resolution equal to that of the SoilGrid data (Dai et al., 2019b; Lu et al., 2020), has been described as "a new version of the global high-resolution dataset of soil hydraulic and thermal parameters for land surface modeling." This dataset is commonly used to provide SHPs for land-surface models (Dai et al., 2019b; He et al., 2020), and is hereafter referred to as DaiYJ. Another key SHPs dataset was developed by Yonggen Zhang, based on a combination of the PTF method and surface soil properties from the SoilGrid dataset. It has a spatial resolution of 1 km × 1 km for the surface layer (0–5 cm, Zhang et al., 2018a), and is hereafter referred to as ZhangGY. This dataset has been described as "a high-resolution global map of soil hydraulic properties produced by a hierarchical parameterization of a physically-based water retention model."

Most of the SHP observations were sampled from the surface layer (uppermost 5 cm) or the subsurface layer (25 cm), and it is therefore robust to assess the accuracy of the soil property datasets for the surface layer, the subsurface layer and for the profile (0–30 cm). However, since the SHPs (the soil saturated hydraulic conductivity) are available for the surface layer (0–5 cm) from the ZhangYG dataset, the SHP observations are used to evaluate for this layer in this datasets.

**Table 1. Calculation of SHPs at different depths (5 cm, 25 cm, 0–30 cm) for evaluating the soil property datasets.**

| Depth | HWSD | SoilGrid | ZhangYG | DaiYJ | observation |
|---|---|---|---|---|---|
| 5 cm | - | $SG_{0-5}$ | $ZhangYG_{0-5}$ | $DaiYJ_{0-5}$ | $obs_5$ |
| 25 cm | - | $SG_{15-30}$ | - | - | $obs_{25}$ |
| 0-30 cm | $HWSD_{0-30}$ | $(5 \cdot SG_{0-5} + 10 \cdot SG_{5-15} + 15 \cdot SG_{15-30})/30$ | - | - | $(obs_5 + obs_{25})/2$ |

Note: *SG* and *obs* represent soil properties from the SoilGrid dataset and from observations, respectively. The subscript gives the soil depths for the SoilGrids properties, and "-" indicates that data for a specific layer are not available in the datasets.

### 2.4.2 Soil moisture datasets

We also used this large-scale observation dataset to evaluate three SM product types that are widely-used for mountainous areas: reanalysis products, data assimilation products and remote-sensing products. The ERA5_Land dataset (the land
component of the fifth generation of the European Reanalysis series) was selected to represent the current state-of-the-art reanalysis product for SM (Muñoz-Sabater et al., 2021). ERA5_Land provides SM data for different soil layers (0–7 cm, 7–28 cm and 28–100 cm) from 1950 to present, with hourly temporal resolution and 9 km spatial resolution. The GLDAS2.1_Noah dataset (from the Noah land surface model, driven by Global Land Data Assimilation System) was taken to represent the newest data-assimilation SM product (Rodell et al., 2004; Beaudoing et al., 2020). GLDAS2.1_Noah provides
SM data for different soil layers (0–10 cm, 10–40 cm and 40–100 cm) from 2000 to present, with spatial and temporal resolutions of 0.25 degrees and three hours, respectively (Rodell et al., 2004). The SMAP_L4 product (Soil Moisture Active Passive L4) is taken as representative of remote-sensing-derived SM products. It provides SM data for two soil layers (0–10 cm and 0–100 cm) from March 2015 to present, with spatial and temporal resolutions of 9 km and three hours, respectively (Reichle et al., 2017).

Our SM observation dataset comprises monthly data at depths of 5 cm, 15 cm, 25 cm, 40 cm and 60 cm for the growing seasons from 2014 to 2020. To use these data for evaluating the differently derived-SM products, we first averaged the derived data to monthly resolution, and then used depth-weighted averaging to convert all data (observations and derived data) to surface (0–10 cm), subsurface (10–100 cm), and profile (0–100 cm) values. SMAP data are only available starting from 2015, thus the assessment was carried out at monthly resolution for data at three depths (surface, subsurface and profile) for the
growing seasons of 2015–2020.

**Table 2. Calculation of the soil moisture at different depths (surface, subsurface and profile) from different datasets.**

| Product | Surface (0-10 cm) | Subsurface (10-100 cm) | Profile (0-100 cm) |
|---|---|---|---|
| GLDAS | $sm_{0\text{-}10}$ | $(30 \cdot sm_{10\text{-}40} + 60 \cdot sm_{40\text{-}100})/90$ | $(10 \cdot sm_{0\text{-}10} + 30 \cdot sm_{10\text{-}40} + 60 \cdot sm_{40\text{-}100})/100$ |
| ERA5_Land | $(7 \cdot sm_{0\text{-}7} + 3 \cdot sm_{7\text{-}28})/10$ | $(18 \cdot sm_{7\text{-}28} + 72 \cdot sm_{28\text{-}100})/90$ | $(7 \cdot sm_{0\text{-}7} + 21 \cdot sm_{7\text{-}28} + 72 \cdot sm_{28\text{-}100})/100$ |
| SMAP_L4 | $sm_{0\text{-}10}$ | $(10 \cdot sm_{0\text{-}100} - sm_{0\text{-}10})/9$ | $sm_{0\text{-}100}$ |
| Observation | $sm_5$ | $(10 \cdot sm_{15} + 10 \cdot sm_{25} + 20 \cdot sm_{40} + 50 \cdot sm_{60})/90$ | $(10 \cdot sm_5 + 10 \cdot sm_{15} + 10 \cdot sm_{25} + 20 \cdot sm_{40} + 50 \cdot sm_{60})/100$ |

Note: $sm$ is soil moisture, and the subscripts indicate the range of depths over which the data are valid. For the derived products, the depth is a range (e.g., 0-10, representing 0–10 cm), while for the observations, $sm$ is reported at the observed depths of 5 cm, 15 cm, 25 cm, 40 cm and 60 cm.

**2.5. Statistic analysis**

The maximum, minimum, mean, and coefficient of variation (CV) were calculated for the SHPs from the different datasets, and a boxplot shows the scatter. The differences between the data in different groups were assessed using a one-way analysis of variance (ANOVA) with the post-hoc Bonferroni test when the normality and homogeneity of variance of the datasets were satisfied. The Kruskal-Wallis ANOVA with a post-hoc Dunn's test was used for cases where these conditions were not satisfied





(McDonald, 2009). The significance level for acceptance was 0.05 for all statistical tests. The statistical analysis and fitting of the different SWRC models to the experiment data were performed using Matlab (R2017b, The MathWorks). The spatial distribution of the data was interpolated using ArcGIS software.

## 3. Results

### 3.1. Basic information of the soil dataset

Some samples were lost while being transported from the field to the laboratory and the soil depth varied across the large-scale study area, meaning that at some random sampling sites we could only sample the uppermost soil layers (0–10 cm), and could only sample the uppermost 0–30 cm of the soil at some long-term measuring stations. In total, we collected 451 disturbed soil samples and 337 undisturbed soil samples from 206 random sampling sites (2 soil depths: 5 cm and and 25 cm) and from 32 long term observation stations (5 soil depths: 5 cm, 15 cm, 25 cm, 40 cm, and 60 cm).

The soil texture for the disturbed samples was classified using the USDA (United States Department of Agriculture) triangle classification system (Figure 3). The main soil types were silt loam (78.05%), sandy loam (12.64%), silt (7.98%), loamy sand (0.89%) and loam (0.44%).

The performances of the different SWRC models for fitting the experimental SWRC data are shown in Figure 4. The results show that the Van Genuchten model fitted the data with significantly greater accuracy (median RMSE and $R^2$ of 0.0027 and
0.999, respectively, at $p < 0.01$ ) than the Kosugi model did (median RMSE and $R^2$ of 0.0038 and 0.998, respectively). The accuracy of the fits made using the Van Genuchten model and Kosugi model were significantly higher than those for the Gadner, Brook-Corey or Clapp and Hornberger models. The least accurat fit was from the Clapp and Hornberger model, with a median RMSE and $R^2$ of 0.0145 and 0.976, respectively. There was no significant difference between the accuracies of the Gadner, Brook-Corey and Clapp and Hornberger models ($p > 0.05$). Thus, the Van Genuchten model was selected as the most
suitable model for the study area, and the soil hydraulic parameters ($\alpha$, $\theta_s$, $\theta_r$ and $l$) were calculated for the samples using this model.

The average and $C_V$ (the standard deviation divided by the average) were calculated for the soil texture (clay, silt, sand), soil bulk density (BD), SOC, $K_S$, and the soil water retention curve parameters ($\alpha$, $\theta_s$, $\theta_r$ and $l$) of Van Genuchten model for all the soil samples at different depths, as shown in Table 3. The results show that $C_V$ of the $n$ (ranged within 0.09–0.12 at different
depths) of Van Genuchten model is less than 0.16, which is a relatively low spatial variability. Meanwhile, $C_V$ for clay ranged from 0.18 to 0.28, and ranged from 0.18 to 0.23 for silt. $\theta_s$ ranged from 0.21 to 0.23, and the soil BD ranged from 0.21 to 0.24. These SHPs therefore show a relatively moderate variability according to Wilding (1985). $C_V$ of sand ranged between 0.45 and 0.62, $\theta_r$ ranged between 0.38 and 0.73, $K_S$ ranged between 1.01 and 1.27, and $\alpha$ ranged from 1.2 to 3.8. The $C_V$ values of these SHPs were higher than 0.36, indicating strong spatial variability. Thus, the soil texture for clay and silt had the lowest
spatial heterology, while both $K_S$ and $\alpha$ had high spatial variability in the study area.

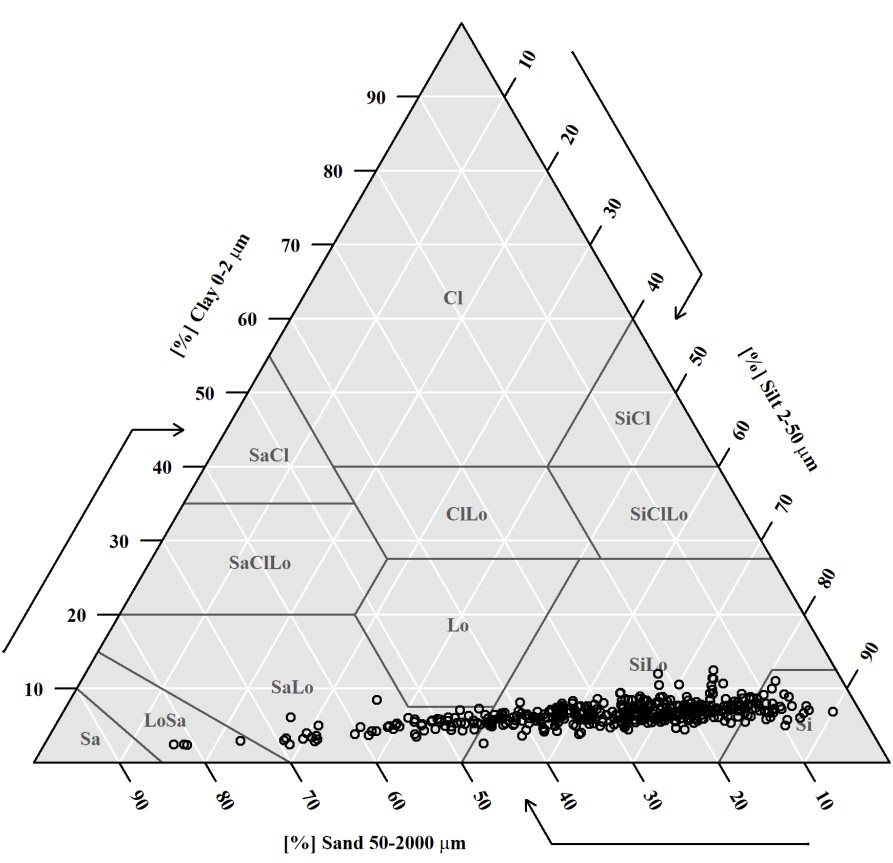

**Figure 3. The distribution of soil textures for all the soil samples in the USDA triangle.**

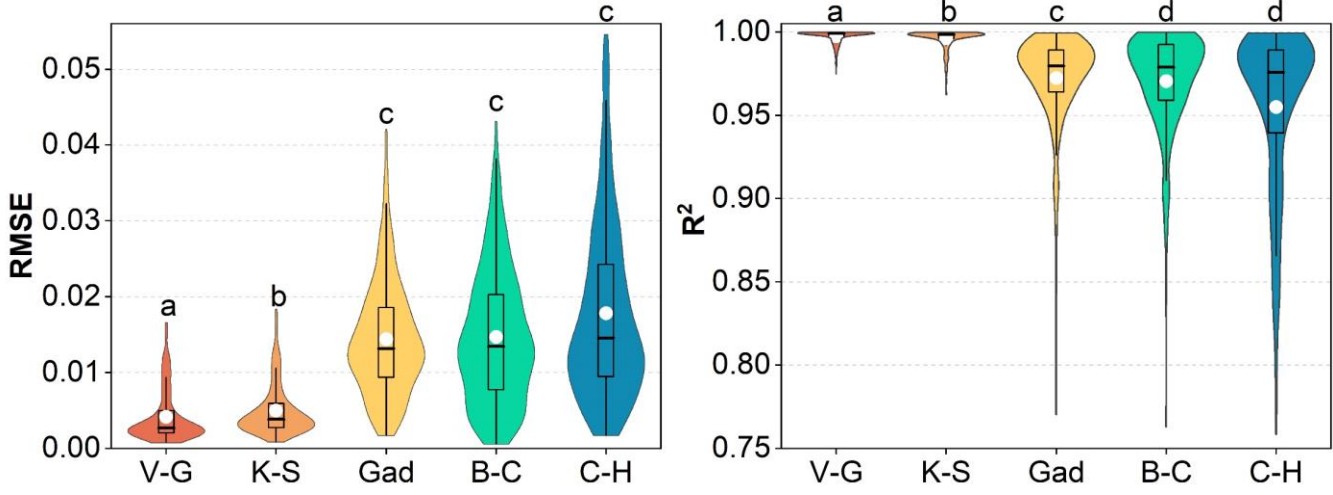

**Figure 4. Comparison of the performance of different SWRC models. The different letters above the voilin plot indicate the**
**significant differences between different models. V-G, K-S, GAD, B-C and C-H represent the Van Genuchten model, Kosugi model,**
**Gadner model, Brook-Corey model and Clapp and Hornberger model, respectively.**

**Table 3.** The descriptive statistics (median and coefficient of variation) for soil properties at different depths.

| depth (cm) | clay (%) | silt (%) | sand (%) | Bulk* (g/cm³) | SOC (%) | $Log_{10}K_S$* (cm/d) | $\alpha$ (cm⁻¹) | $n$ (-) | $\theta_s$* (cm³/cm³) | $\theta_r$ (cm³/cm³) |
|---|---|---|---|---|---|---|---|---|---|---|
| 5 | 6.6 (0.224) | 66.562 (0.192) | 26.686 (0.453) | 1.153 (0.207) | 4.024 (0.889) | 1.445 (0.356) | 0.022 (0.896) | 1.37 (0.119) | 0.55 (0.189) | 0.107 (0.564) |
| 15 | 6.356 (0.156) | 66.607 (0.174) | 26.008 (0.439) | 1.151 (0.195) | 1.466 (1.019) | 1.564 (0.293) | 0.026 (0.877) | 1.382 (0.1) | 0.533 (0.187) | 0.126 (0.472) |
| 25 | 6.381 (0.243) | 65.776 (0.237) | 27.362 (0.513) | 1.189 (0.228) | 2.077 (0.945) | 1.454 (0.381) | 0.026 (0.707) | 1.349 (0.134) | 0.514 (0.214) | 0.094 (0.654) |
| 40 | 6.656 (0.203) | 64.831 (0.222) | 29.213 (0.505) | 1.189 (0.233) | 1.327 (0.998) | 1.436 (0.405) | 0.027 (0.948) | 1.412 (0.117) | 0.462 (0.223) | 0.129 (0.603) |
| 60 | 6.848 (0.287) | 70.089 (0.233) | 20.724 (0.618) | 1.238 (0.238) | 1.240 (1.05) | 1.114 (0.534) | 0.021 (1.094) | 1.394 (0.104) | 0.502 (0.216) | 0.146 (0.544) |
| all | 6.552 (0.227) | 66.323 (0.21) | 26.85 (0.486) | 1.168 (0.219) | 1.473 (1.008) | 1.449 (0.37) | 0.024 (0.887) | 1.374 (0.117) | 0.526 (0.200) | 0.113 (0.566) |

*indicates that the soil property is significantly ($p < 0.05$) different at different depths.

### 3.2 Characteristics of the soil property data

#### 3.2.1 Relationships between different soil properties

After getting the SHPs for the study area, we explored the relationships between different SHPs for the data at each depth, and for the 'all depths' summary data. The summary data included the greatest number of data, and so we used the summary data from all depths as to represent the correlations between the different SHPs (Figure 4). The results show that the relationships between the properties for silt, sand and clay are significant ($p < 0.001$), Pearson's correlation coefficient (R) is -0.99 for the correlation between sand and silt, R is 0.66 for sand and clay, and is 0.60 for silt and clay. We also found that BD is significantly ($p < 0.05$) correlated with soil texture as follows: BD was negatively correlated with clay (R = -0.15) and silt (R = -0.28), while positively correlated with sand (R = 0.28). The relationships between SOC and soil texture (clay, silt and sand) are not significant ($p > 0.05$), while SOC is significantly and negatively correlated with BD (R = -0.72). Similarly, $\log_{10}$ ($K_S$) ($log_{10}$ transformed $K_S$) is not significantly correlated with soil texture, but is negatively correlated with soil BD (R = -0.27) and is significantly positively correlated with SOC (R = 0.18). Finally, the relationships between the four parameters of the Van Genuchten model ($\alpha$, $n$, $\theta$s, $\theta$r) are all significant ,except for the relationship between $n$ and $\theta$s (R = 0.019, $p > 0.05$). The relationships between $\alpha$ and the other SHPs are not significant, except for a significant positive correlation with $K_S$ (R = 0.29). $n$ is negative correlated with clay (R = -0.13), silt (R = -0.19) and SOC (R = -0.29), while significantly positively correlated with sand (R=0.19); $n$ is not significantly correlated with either BD or $K_S$. $\theta$s is positively correlated with silt (R = 0.23) and $K_S$ (R = 0.26), and significantly negatively correlated with sand (R = -0.22) and BD (R = -0.27). $\theta$r is negatively correlated with SOC (R = -0.46) and positively correlated with $K_S$ (R = 0.15), but is not significantly correlated with the other SHPs.



**Figure 5. Correlations between different SHPs.** Different colors indicate different soil layers. The lower triangle of the figure area shows the scatterplots between different SHPs for the different soil layers. The upper triangle lists the Peason's correlation coefficients (R) for the comparisons between the different soil layers, and the first number in each box (Corr: ) is the R value calculated by combinig data from all soil depths. Plots running the diagonally across the figure area represent the distribution of each SHP for the different soil layers.



### 3.2.2. Spatial distribution of soil properties

We explored the spatial distribution of SHPs by using the Kriging method in ArcGIS to interpolate the SHPs that were calculated from the in-situ observations (Figure 6). The results show that clay and silt have a similar spatial distribution, with higher prevalence in the east and lower prevalence in the west and south parts of the study area. This is the opposite of the spatial distribution for sand, which has lower prevalence in the east, and is more common in the west and south parts of the study area. The spatial distribution of BD is similar to that for sand, and opposite to the distribution of clay and silt, which agrees with the correlations found among the SHPs. $K_S$ has higher values in both the middle and north parts of the study area,

and is lowest in the east part of the study area. The residual SM is high in the middle and south parts of the study area, while the saturated SM is high in the southeastern and middle parts of the study area. $\alpha$ has a similar spatial pattern to silt, and is higher in the southeast; $n$ has a similar spatial pattern to the residual soil moisture, with higher values in the middle part of the study area and lower values in the east and west parts.

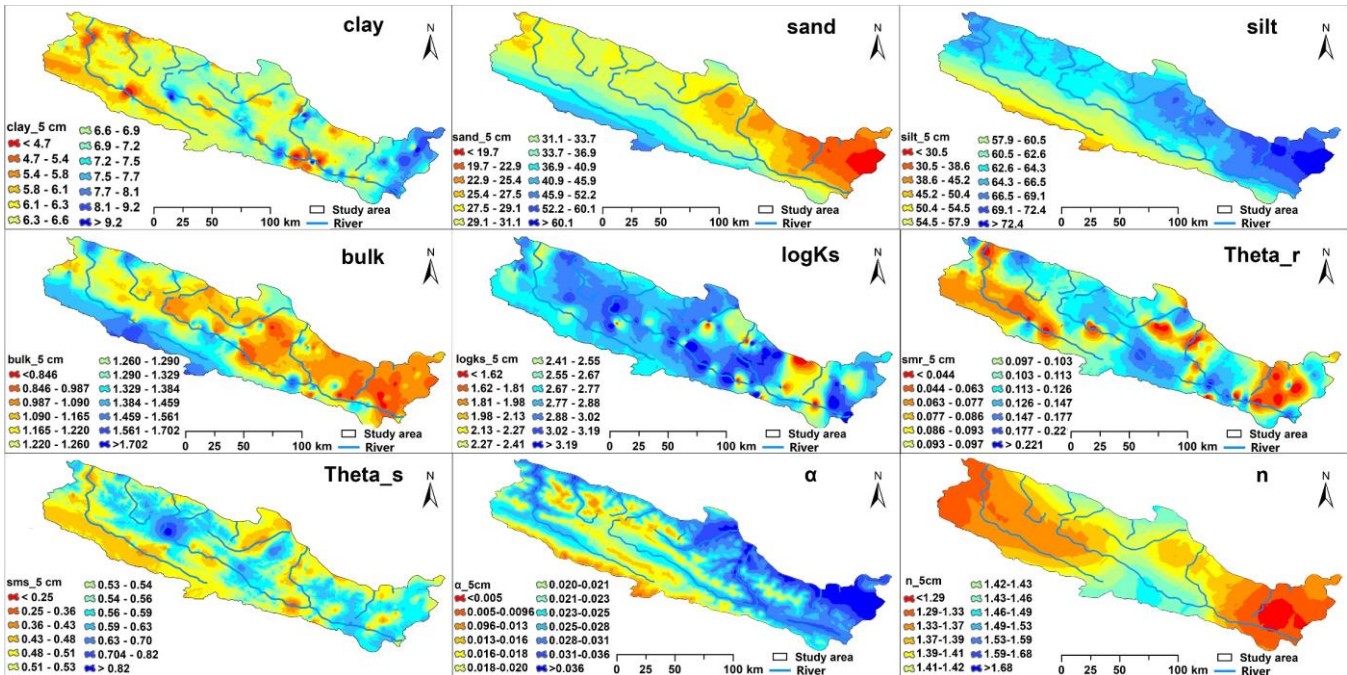

**Figure 6. The spatial distribution of soil texture (sand, silt, clay, %), BD (g/cm³), $\log_{10}(K_S)$ ($\log_{10}$ transformed, cm/d), the residual SM (Theta_r, cm³/cm³), saturated SM (theta_s, cm³/cm³), $\alpha$ and $n$ in the study area.**

### 3.2.3 Vertical distribution of soil properties

We analyzed the vertical distribution of the SHPs in the study area, with the results shown in Table 3 and Figure 7. The soil texture varies as follows: the median clay content decreases from 5 cm depth (where it has a median value of 6.6%) to 15 cm

depth (6.36%), then increases to 60 cm depth (6.85%), while the silt content decreases from 5 cm depth (where it has a median value of 66.56%) to 40 cm depth (64.83%), then increases to 60 cm (70.09%). $\log_{10}K_S$ increases from 5 cm depth (with a



median value of 1.45 cm/day) to 15 cm depth (1.56 cm/day), then decreases between 15 cm and 60 cm depth (1.14 cm/day). The BD increases from 5 cm depth (with a median value of 1.15 g/cm³) to 60 cm depth (1.24 g/cm³). The SOC has a maximum at 5 cm depth (where it has a median value of 4.02%), then increases between 15 cm depth (1.47%) and 25 cm depth (2.08%), and then decreases to 60 cm depth (1.24%). The saturated SM increases from 5 cm depth (0.55 cm³/cm³) to 40 cm depth (0.46 cm³/cm³), then increased from 40 cm to 60 cm depth (0.50 cm³/cm³). The residual SM first increases from 5 cm depth (0.17 cm³/cm³) to 15 cm depth (0.13 cm³/cm³), and then decreases to 25 cm depth (0.094 cm³/cm³), then increases from 25 cm to 60 cm depth (0.15 cm³/cm³). The analysis of variance was used to explore differences in the soil properties at different depths, as shown in Table 3. The results show that differences between most of the SHPs (including clay, silt and sand) at different depths are not significant ($p > 0.05$). However, the differences between some of the SHPs at different depths are significant, including BD ($p = 0.022$), $K_S$ ($p = 0.013$) and saturated SM ($p = 0.022$). The multiple comparison analysis shows that $K_S$ is significantly lower at 60 cm depth than at shallower depths. We should note that there are some differences in the number of samples for different soil layers in this study: we sampled from 206 random sites at depths of 5 cm and 25 cm, and used data from 32 long term monitoring sites that was sampled at depths of 5 cm, 15 cm, 25 cm, 40 cm and 60 cm.

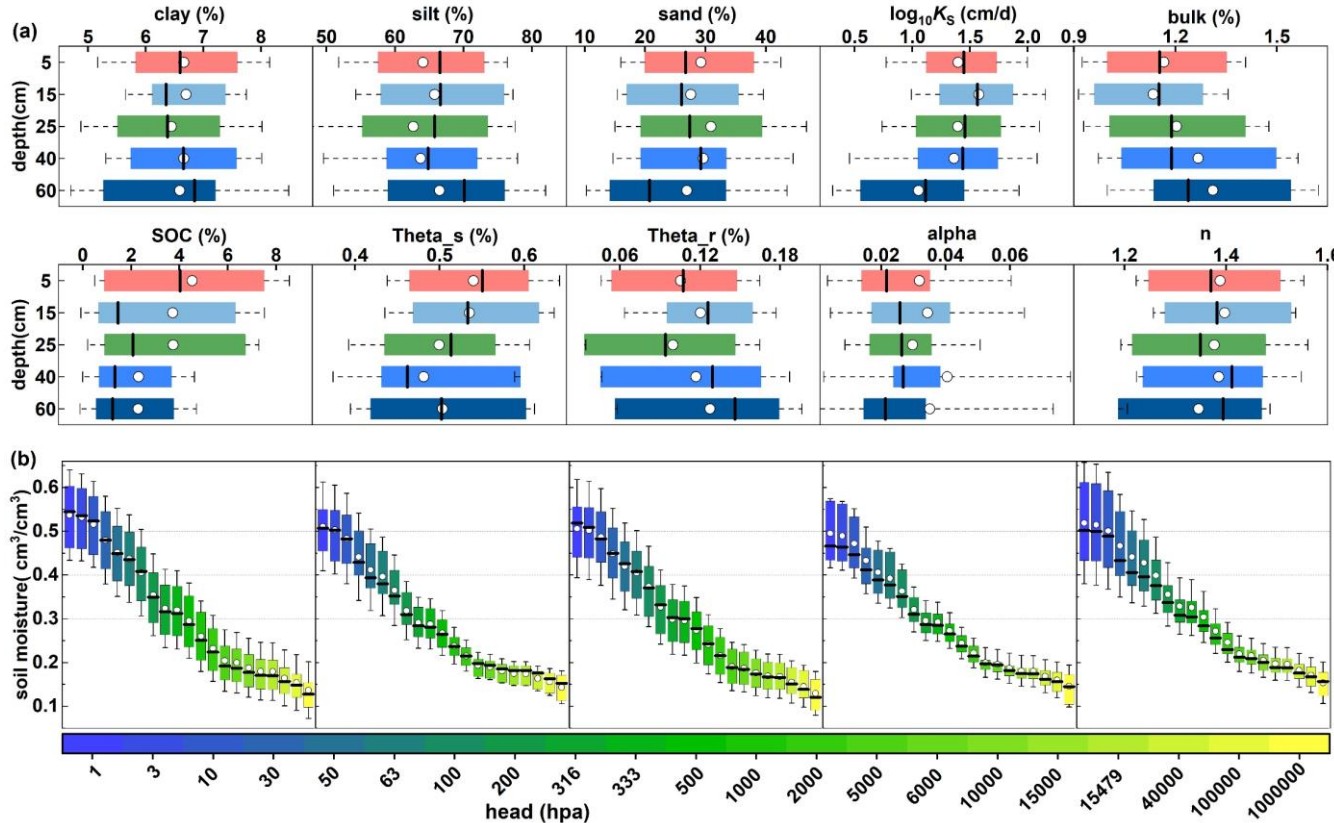

**Figure 7. (a) The vertical distribution of sand, clay, silt, *log₁₀K*ₛ, soil BD, SOC and the parameters of the Van Genuchten model for fitting the soil water retention curve (saturated soil moisture and residual soil moisture (cm³/cm³)) in the study area (32 points); (b) boxplots show the distributions of the soil water retention curves for different soil layers.**





### 3.3 Spatial-temporal characteristics of soil moisture data

The temporal variability of the monthly observed SM data for different soil layers during growing seasons from 2014 to 2020 is shown in Figure 8 (a). Trends in the annual SM observations were explored using linear regression (the slope of the linear regression fitted to data from each station) and are shown in Figure 8 (b). The results show that SM at 5 cm depth followed an increasing trend at 12 of the 32 stations from 2014 to 2020, while there was a decreasing trend at the other 20 stations. For the SM data for all soil layers (0–70 cm) combined, there was an increasing trend at only 7 of the 32 stations, and a decreasing

trend at the other 25 stations. However, the trends at most stations were not significant at the 0.05 level. We calculated the trend for the regionally averaged SM (using an area weight for each SM station) and found that there was a decreasing trend from 2014 to 2020 at all the depths: the slope of lieaner regression is -0.0057 $cm^3 \cdot cm^3$/year, -0.0075 $cm^3 \cdot cm^3$/year, -0.0072 $cm^3 \cdot cm^3$/year, -0.0062 $cm^3 \cdot cm^3$/year, -0.0047 $cm^3 \cdot cm^3$/year, and -0.0094 $cm^3 \cdot cm^3$/year for depths of 5 cm, 15 cm, 25 cm, 40 cm, 60 cm, and for 'full profile SM' (0–70 cm). Meanwhile, the decreasing trend in the area-averaged SM was significant for

depths of 25 cm, 40 cm, 60 cm, and for the 'full profile SM' ($p < 0.05$), but was not significant at 5 cm or 15 cm depth.

The spatial distribution of the average SM during the 2014–2020 growing seasons is shown in Figure 9. The results indicate that SM has a similar spatial pattern at different dpeths, which shows the decreasing trend from the southeast to the northwest of the study area. The SM in the south part of the study area is higher than in the north part, which may be attributable to the relationship between SM and elevation. We analyzed how SM varies with the elevation by comparing the mean observed SM

at each station with the station elevation (Figure S4). This showed that SM increases significantly with increasing elevation ($p < 0.01$), and that the slope of the linear regression between SM and elevation decreases with depth. The slopes were 0.081 ($cm^3 \cdot cm^3$/km), 0.070 ($cm^3 \cdot cm^3$/km), 0.066 ($cm^3 \cdot cm^3$/km), 0.063 ($cm^3 \cdot cm^3$/km), 0.053 ($cm^3 \cdot cm^3$/km), and 0.059 ($cm^3 \cdot cm^3$/km) for SM in the uppermost soil layer (5 cm), in the second layer (15 cm), third layer (25 cm), forth layer (40 cm), fifth layer (60 cm), and in the 'full profile' (0–70 cm), respectively. The vertical varaibility of the SM observations with depth were

also explored using the area-averaged SM from all the stations at each soil depth for each year (Figure 9(f)). The results indicate that SM fluctuations decrease significantly as depth increases ($p < 0.01$). Specifically, SM in the first three layers (the mean SM for 0–25 cm depth is 0.166 $cm^3/cm^3$) was significantly ($p < 0.01$) higher than SM in the fifth layer (the mean SM at 60 cm depth is 0.148 $cm^3/cm^3$), while the SM in the forth layer (the mean SM at 40 cm depth is 0.159 $cm^3/cm^3$) was not significanly different to SM in the other layers.

The relationship between the coefficient of variation ($C_V$) and the mean SM was used to explore spatial-temporal variations in SM over the study area (Figure 10). Figure 11 shows that the $C_V$ decreases as the mean SM increases, and that the relationship between the $C_V$ and mean SM follows a power law. The results indicate that SM is more variable in dry conditions. The spatial variability of SM varies from 0.03 to 0.52 over the study area, with a mean $C_V$ of 0.18, and this is higher than the temporal variability of SM over the study period (2014–2020), which varied from 0.32 to 0.65, with a mean $C_V$ of 0.45.



**Figure 8. (a) The heatmap shows the temporal variability of the observed monthly SM from all stations for 5 soil layers during the growing seasons in 2014–2020 in the study area. (b) The slope of the regression (*k*, cm³•cm⁻³/year) for the annual average SM and time. * and ** show where the slope value for the regression passed the significance test at the 0.05 and 0.01 levels, respectively.**

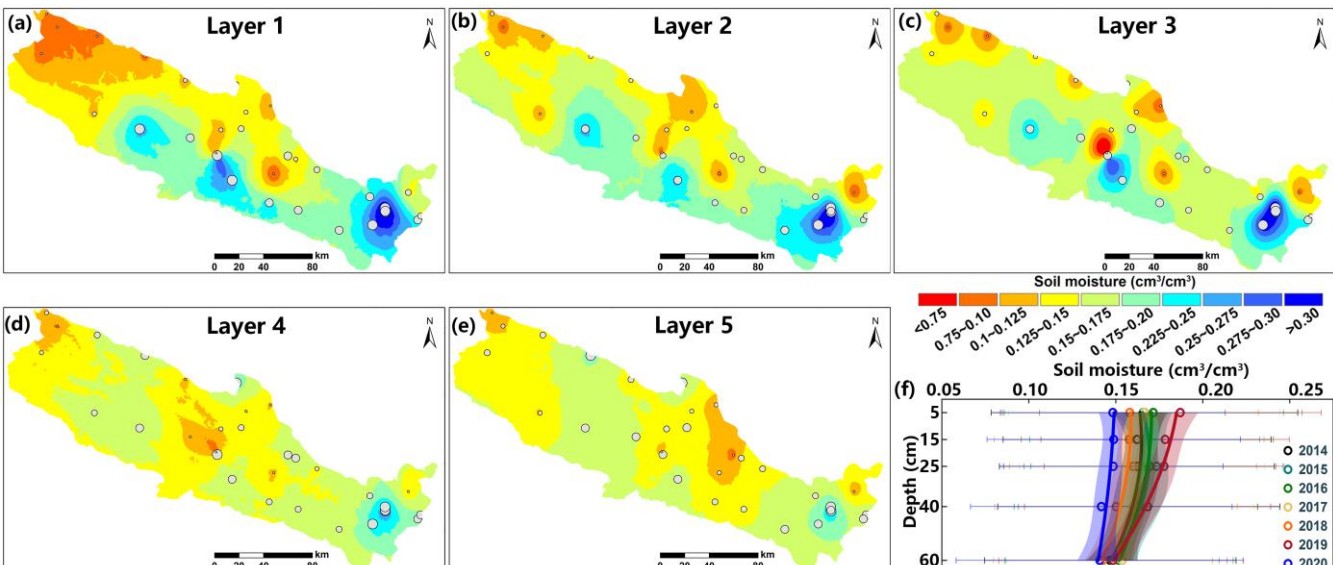

**Figure 9. (a)-(e) The spatial distribution of the average SM during the study period for (a) layer 1 (e) to layer 5. Circles with different sizes show SM measurements from stations with different mean values over the study period. (f) The variability of the average SM with depth in different years. The line and shading show the fitted curve and the 95% confidence interval.**

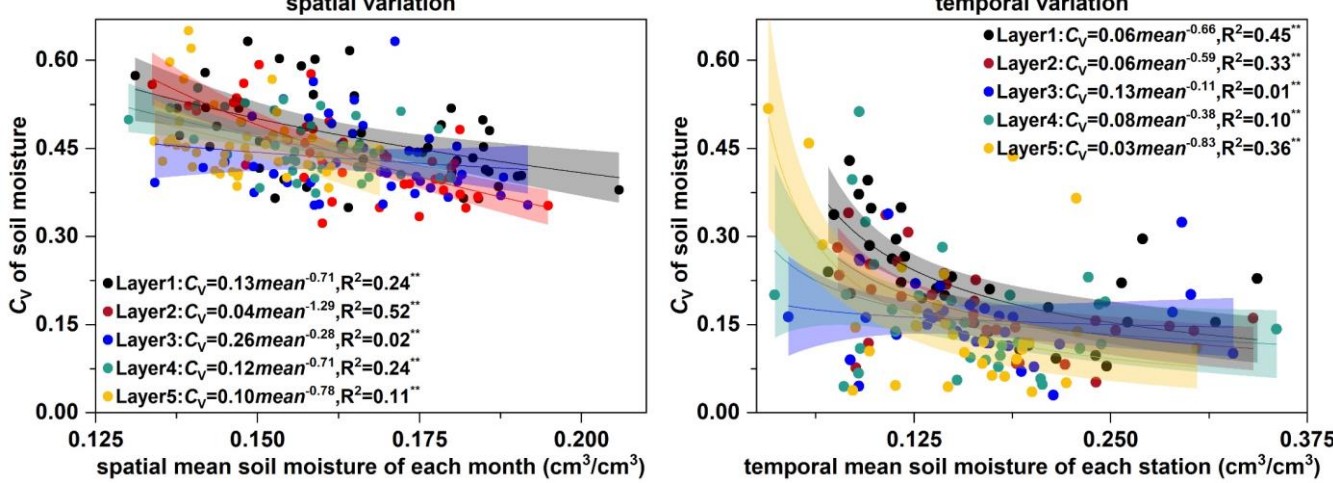

**Figure 10. (a) The relationship between the spatial $C_V$ and mean SM for each station for different soil layers; (b) the relationship**
**between the temporal $C_V$ and mean SM for each month for different soil layers. The curve and shading in each plot show the fitted curve and the 95% confidence interval, and the legend shows the fitting equation between $C_V$ and mean SM for each soil layer. [**] shows where the fitted curves are significant at the 0.01 level.**

### 3.4 Validation of the existing datasets

### 3.4.1 Validation of the existic SHP datasets

Several widely-used derived SHP datasets were evaluated through comparison with the in-situ SHP observations. Based on
the observations, we evaluated the soil texture data (clay, silt and sand) and BD for the SoilGrid and HWSD datasets, and $K_S$





for DaiYJ and ZhangYG datasets (Figure 11). The results show that there is a high positive bias (PBIAS) for clay in the SoilGrid data, with NRMSE (Normalized root mean square error) and PBIAS (Percent bias) of 176% (172%), 203% (199%), and 187% (184%) for depths of 5 cm, 25 cm and 0–30 cm, respectively. The lowest negative biases are for silt in the SoilGrid

dataset, which has an NRMSE (PBIAS) of 42% (-38%), 43% (-35%), and 41% (-36%) for depths of 5 cm, 25 cm and 0–30 cm, respectively. BD was also overestimated in the datasets, with an NRMSE (PBIAS) of 20% (1%), 26% (12%), and 15% (8.7%) for depths of 5 cm, 25 cm and 0-30 cm for the SoilGrid dataset, respectively. Values for $\log K_S$ were also overestimated in the DaiYJ dataset, with an NRMSE (PBIAS) of 36% (11%) and 37% (0.7%) for the depth of 5 cm and 25 cm, respectively. These results suggest that the clay content, sand content and BD are overestimated in both the SoilGrid and HWSD datasets,

and that the silt content is underestimated for 0–30 cm soil depths in the study area. The soil texture and BD values in the SoilGrid dataset have higher precision than those from HWSD, and the $K_S$ values in the ZhangYG dataset have higher precision than those in the DaiYJ for the study area.

Scatterplots comparing the existing soil datasets with the observations are shown in Figure S5. The scatterplots show that the derived and observed values agree most closely for BD, which has the highest R values: 0.30, 0.36 and 0.64 for depths of

5 cm, 25 cm and 0–30 cm, respectively, when comparing the SoilGrid data to the observed data. For 0–30 cm depth, the SoilGrid BD data are more closely to the observations than the HWSD data (R = 0.09). The R for soil texture varies between -0.05 and 0.23 for the SoilGrid and HWSD datasets for different soil depths. R for $K_S$ from the ZhangYG dataset is 0.18 for 5 cm depth, which is higher than R for $K_S$ from the DaiYJ dataset, which is -0.08 and -0.13 for depths of 5 cm and 25 cm, respectively. We also found that SHPs vary within a narrower range in the derived datasets than they do in the observations.

The low Pearson's R value and the scatterplot indicate that the soil datasets do not capture the spatial distribution of the SHPs (soil texture, BD and $K_S$) in mountainous areas with complex terrain.

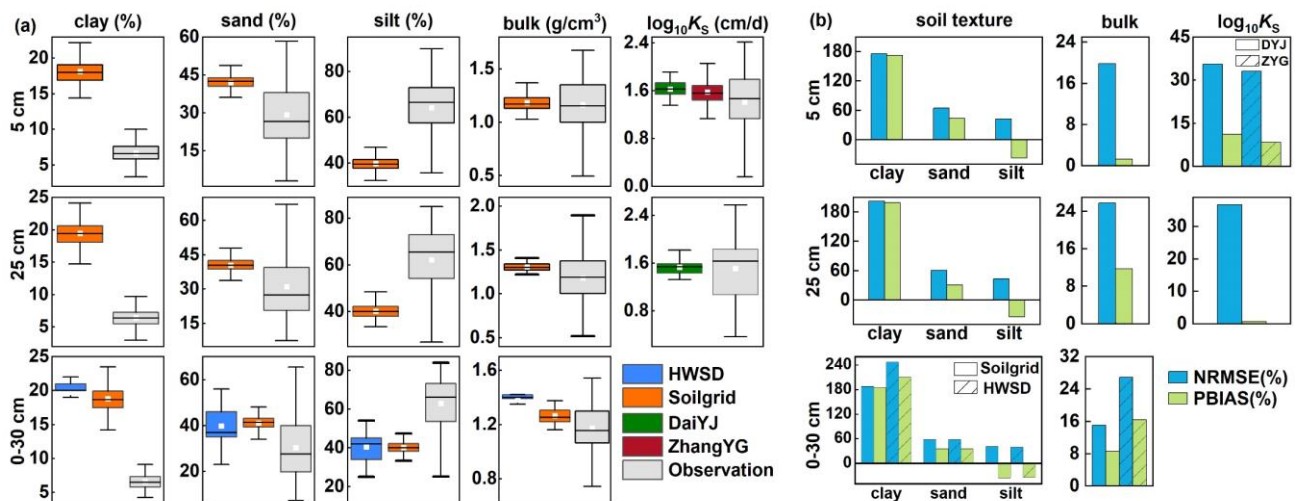

**Figure 11. (a) Boxplots show the distributions for the the evaluated derived datsets (soil texture and BD from the SoilGrid and HWSD datasets, $log_{10}K_S$ from the ZhangYG and DaiYJ datasets), with the corresponding distributions for the observations. (b) Metrics to**

**compare the existing derived soil datasets with the corresponding observations in the study area.**



### 3.4.2 Validation of the soil moisture products

Evaluations of the reanalysis product (ERA_Land), reanalysis product with data assimilation (GLDAS_Noah), and the remote sensing product (SMAP_L4) using the SM observations are shown in Figure 12 (scatterplots comparing the different SM products) and Figure 13 (the results of the evaluation metrics for the different products). The results show that both SMAP_L4

(median RMSEs of 0.039, 0.074 and 0.070 for surface, subsurface and profile soil layers, respectively) and GLDAS_Noah (median RMSEs of 0.068, 0.058 and 0.053 for surface, subsurface and profile soil layers, respectively) have significantly ($p <$ 0.01) lower RMSE than ERA5_Land (median RMSEs of 0.188, 0.196 and 0.195 for surface, subsurface and profile soil layers, respectively). Both SMAP_L4 (median $R = 0.484$, 0.360 and 0.411 for surface, subsurface and profile soil layers, respectively) and ERA5_Land (median RMSE 0.483, 0.317 and 0.341 for surface, subsurface and profile soil layers, respectively) correlate

with the observations with significantly ($p < 0.01$) higher R values than GLDAS_Noah does (with median $R = 0.290$, 0.207 and 0.228 for surface, subsurface and profile soil layers, respectively). The positive mean bias error (MBE) shows that all the derived products overestimate SM for the surface, subsurface and profile soil layers.

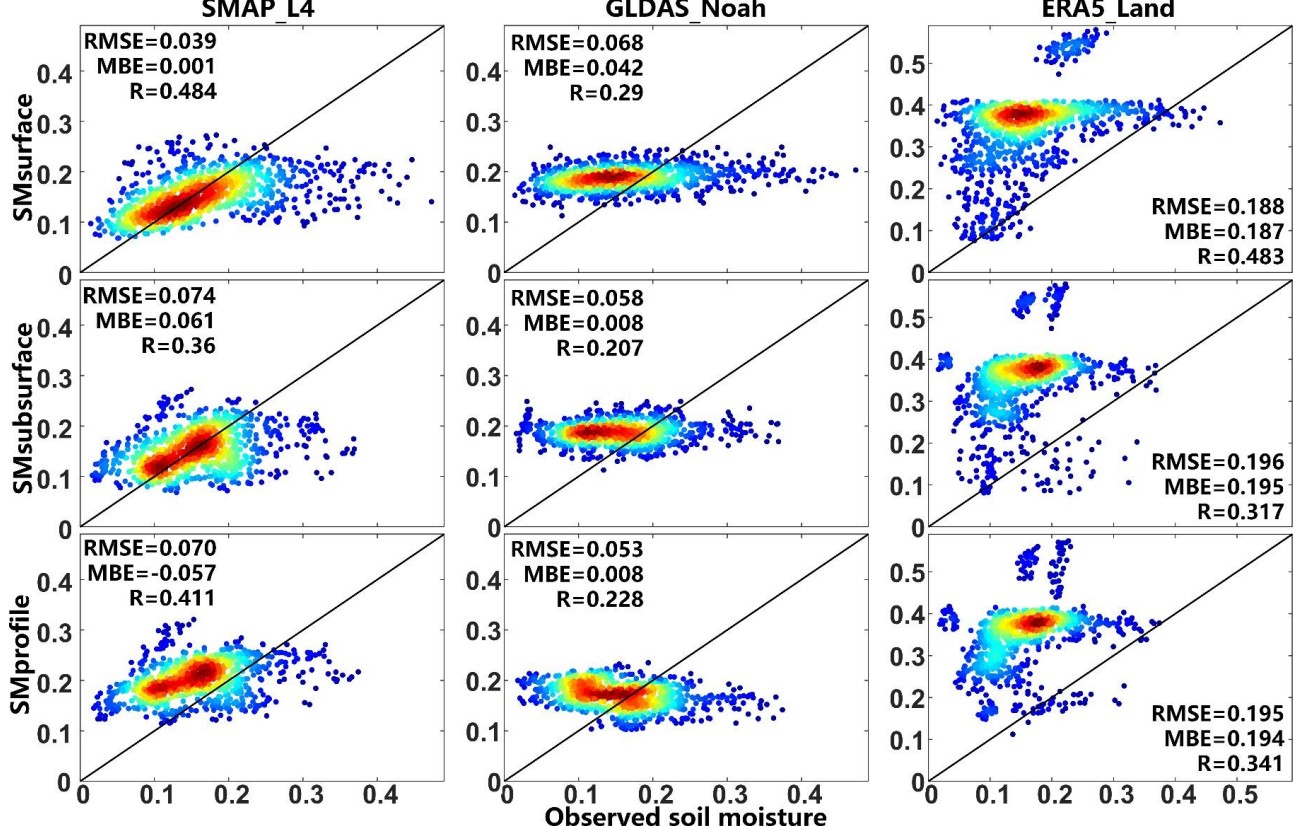

**Figure 12. Scatterplots comparing the different derived SM products with the observed SM for different soil layers. The metrics**
**within each plot show the mean value of the metrics for all stations. The smoothed color density in the scatter plots shows the density of points.**



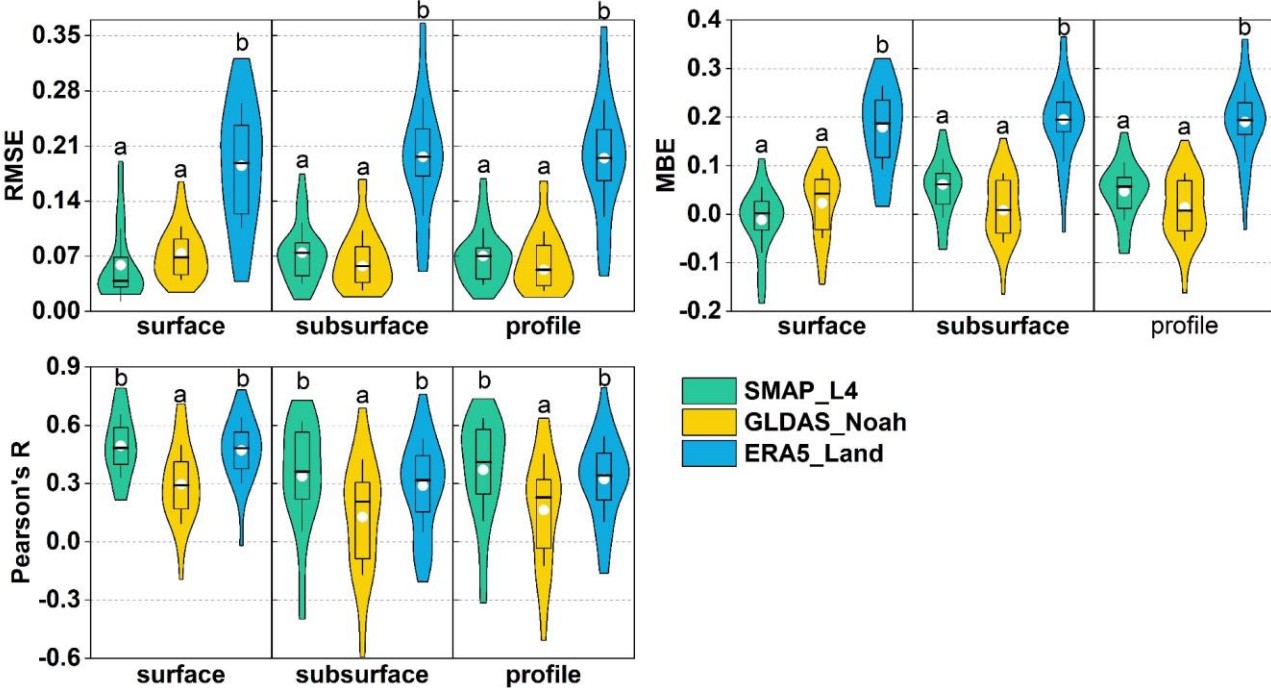

**Figure 13. Metrics for comparsing the different SM products (GLDAS2.1_Noah, ERA5_Land, SMAP_L4) with the in-situ SM observations for different soil layers. The different letters above the volin plot indicate the significant differences (p < 0.05) between different products for each soil layer. The volin curve shows the kernel distribution of the data.**

## 4. Discussion

In this study, we provided a new and unique large-scale SHP and SM dataset for a high and cold mountainous area on the northeastern margin of the Qinghai-Tibet Plateau (QTP), which fills gaps in the spatial coverage of previous studies (Li et al., 2018b; Song et al., 2016; Yang et al., 2016). This dataset provides new knowledge about correlations between different soil properties, and about the spatial and vertical distribution of SHPs, and insights into the spatial-temporal variability of SM over large-scale mountainous areas of the QTP.   The lack of large-scale field sampling has led to our poor understanding of key soil hydraulic properties (including the soil saturated hydraulic conductivity and soil hydraulic properties related to the soil retention curve) over the QTP (Liu et al., 2021; Tian et al., 2017; Zhao et al., 2018). Large-scale information about key soil hydraulic properties is generally estimated using soil pedotransfer functions and basic soil information, such as soil texture (Fatichi et al., 2020), which is strongly location-dependent (Van Looy et al., 2017; Zhao et al., 2020) and not well known over the QTP (Dai et al., 2019; Lu et al., 2020; Zhang et al., 2018a). The large-scale dataset presented here provides comprehensive in-situ observations of both basic SHPs and key soil hydraulic properties. While this study has explored the relationships between different SHPs, further establishing specific pedotransfer function is appropriate over the QTP. Large-scale SM information is commonly taken from reanalysis and remote sensing products, which have high uncertainty (Xia et al., 2014; Xing et al., 2021; Zhang et al., 2019; Zhang et al., 2021). Our SM dataset provides new accurate in-situ SM measurements



covering 2014–2020 over a large-scale mountainous area. It is valuable for future studies into soil hydrological processes (Tian et al., 2019; 2020), for evaluating satellite- and model-based SM products (Zhang et al., 2017; 2019; Su et al., 2020), and for the evaluation of SM-upscaling methods (Jin et al., 2017), and the development of data fusion methods (Lu et al., 2020; Zhang et al., 2018b).

The evaluation of SHPs datasets has shown that some widely-used SHPs datasets (soil texture and BD from the HWSD and SoilGrid datasets, and $K_S$ from the DaiYJ and ZhangYG datasets) have a high bias in the mountainous areas, especially for soil texture (clay and sand contents are significantly overestimated). These derived datasets do not capture the spatial distribution and heterogeneity of SHPs in the mountainous areas, which has been shown to be important in land-processes modelling over the terrain-complex mountainous areas (Jin et al., 2015; Samaniego et al., 2017). The bias and poor spatial
representation of the soil data in these widely used SHPs datasets increase the uncertainty of the land surface models that they are used in, and limit the scientific understanding of and advancement in land surface processes over the QTP (Paniconi and Putti, 2015). Thus, large-scale field soil sampling in high and cold mountainous area is vital to advance large-scale earth system modelling.

    Our evaluation of the three derived SM products indicates that the remote sensing product (SMAP_L4) provides the best
estimates of SM, the reanalysis product (ERA5_Land) does not provide accurate estimates of SM, and the data assimilation product (GLDAS_Noah) fails to capture the temporal variability of SM. A comparison of the ERA5_Land and GLDAS_Noah products indicates that the reanalysis product captures the temporal variability of SM better, while the data assimilation product represents the SM values more accurately. However, remote sensing of large scale SHP and SM remains challenging over the QTP, and such SHP and SM products warranty further validation and improvement over high and cold mountainous areas such
as the QTP. Our SM dataset complements the currently limited in-situ observations and provides opportunity for validating remote sensing SM products, for calibrating land surface models, and for detecting and monitoring land surface changes over QTP.

## 5. Data availability

The dataset of soil hydraulic properties distribution in NetCDF format and soil moisture in excel format is available from
Zenodo repository (https://doi.org/10.5281/zenodo.5830583, He et al., 2022a) and from the National Tibetan Plateau Data Center (https://doi.org/10.11888/Terre.tpdc.271936, He et al., 2022b). The dataset is published under the Creative Commons Attribution 4.0 International (CC BY 4.0) license.



## 6. Conclusions

For this study, in-situ measurements of soil physical properties and a long-term SM monitoring network were set up in a high and cold mountainous area on the northeastern QTP. The resulting dataset presented here fills some geographical gaps in the coverage of SHP data and long-term large-scale SM measurements for the QTP.

Analysis of this in-situ SHPs dataset shows that the spatial distribution and vertical variability of some SHPs vary significantly. SM decreases with increasing depth over 0–70 cm, and SM within 0–70 cm depth shows a decreasing trend for 2014–2020. Evaluations of several existing datasets, using these new measurements, shows that these datasets have a large

bias for soil texture (clay, silt, clay) and do not capture the spatial distribution of SHPs. The SM products based on reanalysis data have a high bias, the SM product based on data assimilation does not capture SM temporal variability, and the SM product based on remote sensing data agrees with the observations more closely than the other datasets do. The observation dataset presented here is significant for assessing uncertainty arising from soil data used in land process models, and for advancing SM-retrieval from remote sensing and land surface models, and in turn, for identifying models and products that are appropriate

for the QTP.

In summary, this study provides a unique and new comprehensive dataset of in-situ measurements of soil physical properties and observations from long-term SM monitoring over the northeastern margin of the QTP. This dataset provides accurate soil parameters and SM evaluation data that are useful for different models, for remote sensing SM products, and for developing and assessing upscaling methods.

**Author contributions**

CH conceptualized and administered the two soil hydraulic property research projects. JT, BZ, CH, conceptualized the methodology. JT and XW collected the data. JT, CH and BZ performed the analysis. JT prepared the draft manuscript and CH revised and finalized the manuscript.

**Competing interests**

The authors declare that they have no conflict of interest.

**Acknowledgements**

The project was partially funded by the National Natural Science Foundation of China (grants 42030501, 91125010, 42101022). We are grateful to the members of the Center for Dryland Water Resources Research and Watershed Science, Lanzhou University, for their persistent efforts to establish and maintain the SM network, collect and analyze the SM data in this high,

cold, and inaccessible mountainous area since 2012. Without their hard work, the dataset presented here would not exist. Members who participate collecting data since 2012 including: Lanhui Zhang, Yibo Wang, Xifeng Zhang, Xin Jin, Weizhen Wu, Jinlin Li, Chen Zhao, Yiwen Jiang, Xiaolei Wang, Lixiao Yang, Xiao Bai, Zhongfu Wang, Xuejin Wang, Yi Zhu, Zhibo



Han, Shengxuan Zeng, Xingyan Tan, Xiang Li, Xuefeng Xu, Feng Li, Yuzuo Zhu, Mingmin Zhang, Xuliang Li, Shaoyuan Xu, Chao Gao, Weijie Hong, Yai Lai, Weiming Kang. We want to thank the Dayekou Hydrology Field Observation and Research Station of Lanzhou University, for their support during the fieldwork. We also thank the high-performance computing service platform in Lanzhou University (https://hpc.lzu.edu.cn/) for providing technical and computing support.

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
