# Peer review of "An in situ observation dataset of soil hydraulic properties and soil moisture in a high and cold mountainous area on the northeastern Qinghai-Tibet Plateau"

_Earth System Science Data, 2022_

## Referee Comment (RC3)

[referee-annotated manuscript omitted]

---

## Author Comment (AC1)

**RC: Reviewer Comment**,   AR: Author Response,   ☐ New Manuscript text

Dear Referee,

We would like to thank you very much for your positive review of our manuscript. Please find our responses to your comments below. These should be considered as preliminary (part of the interactive discussion) as implementation of the final changes also depends on another referee report that is still pending.

Thanks again for your efforts!

Kind regards,

Chansheng He

(on behalf of all the authors)

**The comments of editor and reviewers are in black color with bold text,** the author's answers are indicated in blue color, as well as old text passages. New text passages are indicated in green color.

**General Comments**

**The authors provided datasets of soil properties and long-term soil moisture over the Qilian Mountain based on in-situ observations. The dataset is very useful and important to help the scientific community to understand the soil hydrological processes, to improve the land surface modelling and to develop the soil moisture products over QTP. As the field sampling of profile soil samples and long-term maintenance of soil moisture stations over large scale mountainous areas is difficult, led to the scarce of the large scale in-situ SHP and SM dataset over QTP. Overall, this is a clearly written paper and the structure of the manuscript is well organized. The manuscript can be accepted after addressing my following questions.**

AR: We thank the reviewer for the positive evaluation and the comments.

**Major Comments**

**RC: For the dataset, the spatial distribution of the soil properties datasets has been made public. Besides, it's suggested to upload the representative original measurements of the soil properties (e.g. the key SHP datasets for the main land covers), which can be applied for large-scale modelling and ecohydrological study easily. I also noticed that the number of different soil properties varied at different layers. Please add the detailed instruction of the specific number of each soil properties, which is important for its application.**

AR: As stated in the manuscript, some samples were lost while being transported from the field to the laboratory and the soil depth varied across the large-scale mountainous areas, we finally selected eight sites which has relatively complete profile SHP data and can represent the main land covers (two sites for each main land covers of forestland, high coverage grassland, medium coverage grassland and barren land) in the study area. The selected original measurements of the key SHP datasets for the main land covers have be uploaded into the public datasets (https://doi.org/10.5281/zenodo.6130983). What's more, the specific number of each soil properties (Table S1) has been added in the revised manuscript.

☐ **Table S1. The specific number of the measured soil properties in the datasets.**

| depth/cm | clay | silt | sand | bulk | SOC | $K_S$ | SWRC |
|---|---|---|---|---|---|---|---|
| 5 | 198 | 198 | 198 | 158 | 30 | 174 | 140 |
| 15 | 25 | 25 | 25 | 25 | 25 | 25 | 25 |
| 25 | 162 | 162 | 162 | 64 | 30 | 65 | 56 |
| 40 | 23 | 23 | 23 | 30 | 28 | 30 | 22 |
| 60 | 21 | 21 | 21 | 30 | 24 | 24 | 19 |
| all | 429 | 429 | 429 | 307 | 137 | 318 | 262 |

Note: bulk, SOC, and $K_S$ represent the soil bulk density, soil organic carbon and soil saturated hydraulic conductivity, respectively. SWRC represents the parameters (α, n, saturated soil moisture and residual soil moisture) related to soil water retention curve.

**RC:** **Table1: Why the observed profile SHP is calculated as the average of the surface SHP and subsurface SHP? In my opinion, it should be calculated using the depth-weighting method. The different calculation of profile SHP will influence the validation of profile soil datasets.**

**AR:** Thank you for your reminder. Yes, the profile SHP should be calculated using the depth-weighting method. As the surface depth and subsurface depth of the measured SHP is 5 cm and 25 cm, the surface and subsurface depth represent the depth of 0-15 cm and 15-30 cm, respectively. The profile SHP is calculated as follows:

$$SHP_{profile} = \frac{\left(15 \times SHP_{surface} + 15 \times SHP_{subsurface}\right)}{30} = \frac{\left(SHP_{surface} + SHP_{subsurface}\right)}{2}$$

Thus, the depth-weighted profile SHP is equal to the average of surface and subsurface SHP.

**RC:** **From Figure 6 I can find that the spatial distribution of different soil properties is generated through different method, such as the ordinary Kriging method, the Cokriging method, and the Inverse Distance Weighted method. In my opinion, the different spatial pattern of soil properties can also be caused by the different methods. What's more, what's the spatial resolution of the soil properties dataset?**

**AR:** After evaluating the suitable method of spatial interpolation for each SHPs, the Kriging and Cokriging methods were found to be not suitable for some variables. Thus, the Inverse Distance Weighted method was adopted to generate the measured SHP products. The new figure is showed as Figure 6. And the text about the spatial distribution of SHP, which is different with the previous description, has been changed in the revised manuscript. What's more, the spatial resolution of the generated SHP products is 900 m.

☐ The residual SM is high in the middle and south parts of the study area, while the saturated SM is high in the southeastern and middle parts of the study area. $\alpha$ is high in the southeast and low in the midle part of the study area;

[Figure]

Figure 6. The spatial distribution of soil texture (sand, silt, clay, %), Bulk (g/cm$^3$), $log_{10}K_S$ ($log_{10}$ transformed saturated hydraulic conductivity, cm/day), the residual SM (Theta_r, cm$^3$/cm$^3$), saturated SM (Theta_s, cm$^3$/cm$^3$), $\alpha$ and $n$ in the study area.

**RC: Please notice that the spatial resolution of different SHP datasets or SM datasets are different, which will influence your validation. Please discuss it in the manuscript. Besides, the authors are suggested to validate the three SM products at daily scale instead of monthly scale.**

AR: Thank you for the comments. Yes, the spatial resolutions of the SHP datasets and SM datasets are different. For the SHP datasets, the spatial resolution of HWSD, ZhangYG, and DaiYJ (with spatial resolution of almost 1 km, FAO/IIASA/ISRIC/ISS-CAS/JRC, 2012; Zhang et al., 2018; Dai et al., 2019a) is larger than that of the SoilGrids2.0 (250 m, Poggio et al., 2021). For the SM datasets, the spatial resolution of the GLDAS_Noah product (nearly 25 km, Roddell and Beaudoing, 2017) has a larger spatial resolution than the ERA5_Land and SMAP_L4 product (9 km, Muñoz-Sabater et al.,

2021; Reichle et al., 2017). In our opinion, the different spatial resolutions will influence the spatial heterogeneity of the estimated SHP or SM when comparing with the observations. Thus, the discussion about the influence of spatial resolution on the evaluation of SHP and SM products has been added in the discussion part as follows:

☐ Since the estimated soil properties used the average value to represent soil variable for each grid and neglected the soil spatial heterogeneity within each grid (Dai et al., 2019b; Zheng et al., 2015), the spatial heterogeneity of SHP has been shown to decrease with the increase of spatial resolution (Jin et al., 2015; Lin et al., 2006; Li et al., 2021). It should be noted that the spatial heterogeneity of the estimated SHPs from different soil datasets is also influenced by the spatial resolution of the soil datasets, which leads to the relatively lower R value in the evaluation of the SHP datasets.

We should also note that the performances of GLDAS and ERA5_Land products can also be influenced by its spatial resolution. Since the ERA5_Land product (9 km, Muñoz-Sabater et al., 2021) has the finer spatial resolution than the GLDAS_Noah product (nearly 25 km, Roddell and Beaudoing, 2017), the ERA5_Land product can better capture the spatial distribution pattern of the observed soil moisture than the GLDAS_Noah.

AR: Besides, the validations of the SM products at daily scale was performed as suggested. Results are shown in Figure S6 (scatterplots comparing the different SM products with observations) and Figure S7 (the results of the evaluation metrics for the different products). Results of evaluations indicate that the performance of different SM products at daily scale are similar to that at monthly scale. This part has been added in the revised supplement. The new figures (Figure S6 and Figure S7) are showed as follows:

[Figure]

Figure S6. Scatterplots comparing the different derived SM products with the observed SM for different soil layers at daily scale. The metrics within each plot show the median value of the metrics for all stations. The smoothed color density in the scatter plots shows the density of points.

[Figure]

Figure S7. Metrics for comparing the different SM products (GLDAS2.1_Noah, ERA5_Land, SMAP_L4) with the in-situ SM observations for different soil layers at daily scale. The different letters above the volin plot indicate the significant differences (p < 0.05) between different products for each soil layer.

**Specific Comments**

**RC: L11: Please change the "describing and predicting" to "describe and predict"**

AR: Yes, it was done as suggested.

**RC: L18: Please delete the "of" before "SM".**

AR: Yes, it was done as suggested.

**RC: L35: Please change the "Earth" to "earth".**

AR: Yes, it was done as suggested.

**RC: L41: Please change the "soil-sampling" to "soil sampling"**

AR: Yes, it was done as suggested.

**RC: L42-43: Please rewrite the sentence.**

AR: Yes, the sentence was changed as follows.

☐   This makes the estimated SHP data from many widely-used soil databases highly uncertain for mountainous areas.

**RC: L50: I think "and" should be replaced by "especially".**

AR: Yes, it was done as suggested.

**RC: L51: Please use the abbreviation "SHPs" replace the "soil hydraulic properties"**

AR: Yes, it was done as suggested.

**RC: L52: I think that "individual" should be replaced by "different".**

AR: Yes, it was done as suggested.

**RC: L60-61: The statement is unclear, please rewrite the sentence.**

AR: Yes, the sentence was changed as follows.

☐   Ground-based SM measurements are the most accurate and important means for developing and validating the spatially-contiguous data derived from satellites or from land-surface models

**RC: L68-69: The statement is unclear, please rewrite the sentence.**

AR: Yes, the sentence was changed as follows.

☐   This study presents a long-term SM dataset collected through a SM monitoring network that was established in 2013, and the SHP dataset through field-sampling from both the random soil profiles and the long-term SM monitoring stations, on the northeastern edge of the QTP.

**RC: L72: Please change the "soil-property" to "soil property".**

AR: Yes, it was done as suggested.

RC: **L83: Please change the "land-cover" to "land cover".**

AR: Yes, it was done as suggested.

RC: **L95: Please check the name of the dataset, we can't find the dataset in the website.**

AR: The name is "Landuse/landcover data of the Heihe River Basin (2011)". It was changed as suggested.

RC: **L99: please delete the "mountainous"**

AR: Yes, it was done as suggested.

RC: **L106: It should be "study area"**

AR: Yes, it was done as suggested.

RC: **L109: Please change the "long-term" to "long term"**

AR: Yes, it was done as suggested.

RC: **L115: "Since the soil freezes in winter, SM data are only available for the growing seasons (May to October, Tian et al., 2019)". Why only the SM data are only available for the growing seasons? I think the ECH2O 5TE probe can measure the liquid soil water content during winters.**

AR: Yes, the 5 TE probe can only measure the liquid water content and can't measure the soil water content in solid state (Cobos and Chambers, 2010; Zhao et al., 2013). Most of the soil water converts from the liquid state into solid state during the freezing period in the QTP(Zhao et al., 2013; Zheng et al., 2018; 2019), which leads to the underestimate of the soil water content by the 5TE probe during the winters (Tian et al., 2019; 2020). Thus, we only keep the measured SM data during the growing season to represent the actual soil moisture content in the study area.

RC: **L122: Please delete the "a" before metal cylinder.**

AR: Yes, it was done as suggested.

**RC: L131: You have mentioned the size of the cylinder above, no need to mention it again.**

AR: Yes, it was done as suggested.

**RC: L145: Please delete "," after "(also written as cm H2O)"**

AR: Yes, it was done as suggested.

**RC: L171: What is the specific depth of the surface layer and subsurface layer?**

AR: The depth of the surface layer and subsurface layer is 5 cm and 25 cm, respectively.

**RC: L178: Please delete "types" after product.**

AR: Yes, it was done as suggested.

**RC: L201: Please unify the format of "CV" in the manuscript.**

AR: Thank you. It was unified into "$C_V$" throughout the manuscript as suggested.

**RC: L228: The parameter "I" should be n.**

AR: Yes, it was done as suggested.

**RC: Table3: What is $log_{10}K_S$, please explain.**

AR: $log_{10}K_S$ is the $K_S$ value with the $log_{10}$ transformation. It was illustrated in the revised manuscript.

**RC: L248: It should be Figure5, not Figure4.**

AR: Yes, it was done as suggested.

**RC: Figure7: The unit of bulk is wrong, please correct it.**

AR: Thank you. The unit of bulk has been changed to $g/cm^3$.

**RC: L314: Please change the "full profile SM" to "profile SM"**

AR: Yes, it was done as suggested.

**RC: L315: Please change the "or" to "and"**

AR: Yes, it was done as suggested.

**RC: L311: It's Figure 10 not Figure 11.**

AR: Yes, it was done as suggested.

**RC: L333-334: Please check the value of spatial CV and temporal CV.**

AR: Thank you. It was changed as follows.

☐ The spatial variability of SM varies from 0.32 to 0.65 over the study area, with a mean $C_V$ of 0.45, and this is higher than the temporal variability of SM over the study period (2014–2020), which varied from 0.03 to 0.52, with a mean $C_V$ of 0.18

**RC: Figure 9: Please check the legend of the soil moisture value.**

AR: Thank you. The range of the first-class soil moisture is <0.075, not <0.75. It was changed.

**RC: L353: What is the equation of NRMSE?**

AR: NRMSE is the Normalized root mean square error, it is calculated as follows:

$$NRMSE = 100 \times \sqrt{\frac{1}{N}\sum_{i=1}^{N}(y_i - \hat{y}_i)^2}\Big/\bar{y}$$

Where $y$ and $\hat{y}$ is the observations and simulations of products, respectively. $\bar{y}$ is the mean of observations. $N$ is the number of comparisons.

**RC: L359-362: It's better to move this sentence to the end of next paragraph.**

AR: Thank you. The sentence was moved into the next paragraph and revised as follows.

☐ These results suggest that the clay content, sand content and BD are overestimated in both the SoilGrid and HWSD datasets, and that the silt content is underestimated for 0–30 cm soil depths in the study area. The soil texture and BD values in the SoilGrid dataset have higher precision than those from HWSD, and the $K_S$ values in the ZhangYG dataset have higher precision than those in

the DaiYJ for the study area. Besides, the low Pearson's R value and the scatterplot indicate that the soil datasets do not capture the spatial distribution of the SHPs (soil texture, BD and KS) in mountainous areas with complex terrain.

**RC: L361: What's the performance of ZhangYG dataset?**

AR: Thank you. The estimated $log_{10}K_S$ in the ZhangYG dataset has an NRMSE (PBIAS) and R of 33.1% (8.3%) and 0.17 for the depth of 5 cm in the study area. It was added in the revised manuscript as follows.

☐  While the estimated $log_{10}K_S$ in the ZhangYG dataset has an NRMSE (PBIAS) and R of 33.1% (8.3%) and 0.17 for the depth of 5 cm in the study area.

**RC: Figure 12: Please add the unit of the soil moisture**

AR: The unit is $(cm^3/cm^3)$. It was added as suggested.

**RC: L401: Please check the format here.**

AR: Thank you. We have checked the format, the blank was deleted in the revised manuscript.

**RC: L421: Please delete the "of" after "understanding".**

AR: Yes, it was done as suggested.

**RC: L428-L429: This paragraph is the discussion about soil moisture, please delete the "SHP"**

AR: Yes, it was done as suggested.

**RC: L442: Please change the "some" to "different".**

AR: Yes, it was done as suggested.

References:

Cobos, D. R., and Chambers, C.: Calibrating ECH2O soil moisture sensors, Application Note, Decagon Devices, Pullman, Washington, 2010.

Dai, Y., Shangguan, W., Wei, N., Xin, Q., Yuan, H., Zhang, S., Liu, S., Lu, X., Wang, D., and Yan, F.: A review of the global soil property maps for Earth system models, SOIL, 5, 137-158, https://doi.org/10.5194/soil-5-137-2019, 2019b.

Dai, Y., Xin, Q., Wei, N., Zhang, Y., Shangguan, W., Yuan, H., Zhang, S., Liu, S., and Lu, X.: A Global High-Resolution Data Set of Soil Hydraulic and Thermal Properties for Land Surface Modeling, J. Adv. Model. Earth Sy., 11, 2996-3023, https://doi.org/10.1029/2019MS001784, 2019a.

FAO, IIASA, ISRIC, ISSCAS, and JRC: Harmonized World Soil Database (HWSD), version 1.1, 2012.

Jin, X., Zhang, L. h., Gu, J., Zhao, C., Tian, J., and He, C. S.: Modeling the impacts of spatial heterogeneity in soil hydraulic properties on hydrological process in the upper reach of the Heihe River in the Qilian Mountains, Northwest China, Hydrol. Processes, 29, 3318-3327, https://doi.org/10.1002/hyp.10437, 2015.

Li, X., Xu, X., Wang, X., Xu, S., Tian, W., Tian, J., and He, C.: Assessing the Effects of Spatial Scales on Regional Evapotranspiration Estimation by the SEBAL Model and Multiple Satellite Datasets: A Case Study in the Agro-Pastoral Ecotone, Northwestern China, Remote Sens., 13, 1524, 2021.

Lin, H., Bouma, J., Pachepsky, Y., Western, A., Thompson, J., van Genuchten, R., Vogel, H.-J., and Lilly, A.: Hydropedology: Synergistic integration of pedology and hydrology, Water Resour. Res., 42, https://doi.org./doi:10.1029/2005WR004085, 2006.

Muñoz-Sabater, J., Dutra, E., Agustí-Panareda, A., Albergel, C., Arduini, G., Balsamo, G., Boussetta, S., Choulga, M., Harrigan, S., Hersbach, H., Martens, B., Miralles, D. G., Piles, M., Rodríguez-Fernández, N. J., Zsoter, E., Buontempo, C., and Thépaut, J. N.: ERA5-Land: a state-of-the-art global reanalysis dataset for land applications, Earth Syst. Sci. Data, 13, 4349-4383, https://doi.org./10.5194/essd-13-4349-2021, 2021.

Poggio, L., de Sousa, L. M., Batjes, N. H., Heuvelink, G. B. M., Kempen, B., Ribeiro, E., and Rossiter, D.: SoilGrids 2.0: producing soil information for the globe with quantified spatial uncertainty, SOIL, 7, 217-240, https://doi.org/10.5194/soil-7-217-2021, 2021.

Reichle, R. H., Lannoy, G. J. M. D., Liu, Q., Ardizzone, J. V., Colliander, A., Conaty, A., Crow, W., Jackson, T. J., Jones, L. A., Kimball, J. S., Koster, R. D., Mahanama, S. P., Smith, E. B., Berg, A., Bircher, S., Bosch, D., Caldwell, T. G., Cosh, M., González-Zamora, Á., Collins, C. D. H., Jensen, K. H., Livingston, S., Lopez-Baeza, E., Martínez-Fernández, J., McNairn, H., Moghaddam, M., Pacheco, A., Pellarin, T., Prueger, J., Rowlandson, T., Seyfried, M., Starks, P., Su, Z., Thibeault, M., Velde, R. v.

d., Walker, J., Wu, X., and Zeng, Y.: Assessment of the SMAP Level-4 Surface and Root-Zone Soil Moisture Product Using In Situ Measurements, J. Hydrometeorol., 18, 2621-2645, https://doi.org./10.1175/jhm-d-17-0063.1, 2017.

Roddell, M. and Beaudoing, H.: GLDAS Noah Land Surface Model L4 3 hourly 0:25 × 0:25 degree V2.1, NASA GESDISC DATA ARCHIVE, Greenbelt, Maryland, USA, https://doi.org/10.5067/E7TYRXPJKWOQ, 2017.

Zhang, Y., Schaap, M. G., and Zha, Y.: A High-Resolution Global Map of Soil Hydraulic Properties Produced by a Hierarchical Parameterization of a Physically Based Water Retention Model, Water Resour. Res., 54, https://doi.org/10.1029/2018WR023539, 2018.

Zhao, C.L., Jia, X.X., Shao, M. A., and Zhang, X.: Using pedo-transfer functions to estimate dry soil layers along an 860-km long transect on China's Loess Plateau, Geoderma, 369, 114320, https://doi.org/10.1016/j.geoderma.2020.114320, 2020.

Zheng, D., van der Velde, R., Su, Z., Wang, X., Wen, J., Booij, M. J., Hoekstra, A. Y., and Chen, Y.: Augmentations to the Noah Model Physics for Application to the Yellow River Source Area. Part I: Soil Water Flow, J. Hydrometeorol., 16, 2659-2676, https://doi.org./10.1175/jhm-d-14-0198.1, 2015.

Zheng, D., van der Velde, R., Su, Z., Wen, J., Wang, X., and Yang, K.: Impact of soil freeze-thaw mechanism on the runoff dynamics of two Tibetan rivers, J. Hydrol., 563, 382-394, https://doi.org./10.1016/j.jhydrol.2018.06.024, 2018.

Zheng, D., Li, X., Wang, X., Wang, Z., Wen, J., van der Velde, R., Schwank, M., and Su, Z.: Sampling depth of L-band radiometer measurements of soil moisture and freeze-thaw dynamics on the Tibetan Plateau, Remote Sens. Environ., 226, 16-25, https://doi.org./10.1016/j.rse.2019.03.029, 2019.

---

## Author Comment (AC2)

**RC: Reviewer Comment**,    AR: Author Response,    ☐ New Manuscript text

Dear Referee,

We would like to thank you very much for your effort in reviewing our manuscript. Please find our responses to your comments below. These should be considered as preliminary (part of the interactive discussion) as implementation of the final changes also depends on another referee report that is still pending.

Kind regards,

Chansheng He

(on behalf of all the authors)

**The comments of editor and reviewers are in black with bold text,** the author's answers are indicated in blue color, as well as old text passages. New text passages are indicated in green color.

**General Comments**

**Overall, the paper is well written, and the data are valuable. However, I still have some main concerns:**

AR: We thank the reviewer for the evaluation and the comments.

**Major Comments**

**RC: The authors give a good description of the SHP and SM datasets. However, they fail to clarify the accuracies of the ground-based datasets. Errors exist in either ground measurements or products. Direct comparisons between ground measurements and products can not help us understand the quality of the ground measurements. Although it is still challenging to quantify errors within the in-situ measurement, methods like triple collocation do exist that can give uncertainties of the in-situ measurements. I would strongly recommend the authors to try to explain the**

**accuracy of the ground measurements, at least to cite some previous validation work to prove that the validation results of SHP and SM products in this work are consistent with them. This will make the quality of the ground measurements to be convincing.**

AR: Thank you for your comments. Yes, errors exist in either ground measurements or products. Firstly, we have tried to collect SM using two sensors at the same depth at a station within the study area (Figure R1 (a)). However, as a distance needs to be maintained between the two sensors to prevent interference (Cobos and Chambers, 2010), and SM has strong spatial heterology even at centimeter scale (Liang et al., 2011; Zhang et al., 2018; Guo et al., 2019; Guo et al., 2020), it's difficult to say that the two sensors are measuring SM at the same place. Results also showed that the two sensors at the same depth have different SM dynamics (Figure R1 (b)). Thus, the triple collocation will introduce additional errors when obtaining the local SM. Secondly, the harsh mountainous environment makes it difficult to collect data and maintain the large-scale SM network over QTP. What's more, we usually need to replace several SM sensors each year, which are always broken by the animals (rat, yak, etc.) during the long period (Figure R2). The limited funding also makes it difficult to maintain the long-term SM monitoring over large scale of QTP using the triple collocation method.

[Figure]

Figure R1. (a) The installation of two 5TE sensors at the same layer at one site within the study area, (b) the comparison of measured SM during 2021/5/1-2021/10/15 at the first four depths.

[Figure]

Figure R2. (a) Datalogger damage caused by water ingress, (b) the stolen of 5TE sensor and datalogger in the field (only the broken white waterproof box remains), (c)-(f) the damage of 5TE sensor in the field.

Meanwhile, we have checked our results thoroughly and corrected a mistake when evaluating the SMAP_L4 product in our previous analysis (the SMAP_L4 product version during 2015-2016 is different from the data product after 2016). What's more, in order to compare our results with the previous evaluations, we evaluated the SM products using the metrics of Pearson's R, bias, and ubRMSE (Figures 12 and 13). The results about evaluations of the SM products have been revised in the manuscript accordingly.

[Figure]

Figure 12. Scatterplots comparing the different derived SM products with the observed SM for different soil layers. The metrics within each plot show the mean value of the metrics for all stations.

[Figure]

Figure 13. Metrics for comparing the different SM products (GLDAS_Noah, ERA5_Land, and SMAP_L4) with the in-situ SM observations for different layers. The different letters above the violin plot indicate the significant differences ($p < 0.05$) between different products for each soil layer, while no letter indicates the differences are not significant.

☐ The results show that both SMAP_L4 (mean bias of -0.011 $cm^3/cm^3$, 0.052 $cm^3/cm^3$, and 0.045 $cm^3/cm^3$ for surface, subsurface, and profile soil layers, respectively) and GLDAS_Noah (mean bias of 0.023 $cm^3/cm^3$, 0.013 $cm^3/cm^3$, and 0.014 $cm^3/cm^3$ for surface, subsurface, and profile soil layers, respectively) have significantly ($p < 0.01$) lower bias than ERA5_Land (mean bias of 0.179 $cm^3/cm^3$, 0.191 $cm^3/cm^3$, and 0.190 $cm^3/cm^3$ for surface, subsurface, and profile soil layers, respectively). Both SMAP_L4 (mean R of 0.490, 0.343, and 0.371 for surface, subsurface, and profile soil layers, respectively) and ERA5_Land (mean R of 0.471, 0.289, and 0.323 for surface, subsurface, and profile soil layers, respectively) correlate with the observations with significantly ($p < 0.01$) higher R values than GLDAS_Noah does (with mean R of 0.296, 0.127, and 0.164 for surface, subsurface, and profile soil layers, respectively). After removing the bias, the mean ubRMSE of the three SM products are lower than 0.032 $cm^3/cm^3$, 0.024 $cm^3/cm^3$, and 0.024 $cm^3/cm^3$ for the surface, subsurface, and profile

SM, respectively. Therefore, the three SM products achieved the accuracy requirement of 0.04 cm$^3$/cm$^3$ (Chan et al., 2016) in the study area.

After summarizing the previous works about SM products validation, we found that our results are consistent with the previous studies. For example, Xing et al. (2021) found that for the surface SM, SMAP_L4 has a better performance than ERA5_Land with the higher R and lower bias. For the root zone SM, SMAP_L4 and ERA5_Land have higher R values than the GLDAS_Noah, while SMAP_L4 and GLDAS_Noah have a lower bias than the ERA5_Land (Xing et al., 2021), which are consistent with our results. On the other hand, previous research also displayed a broad range of bias, ubRMSE, and R when evaluating the three SM products (Table R1, Bi et al., 2016; Qu et al., 2019; Xing et al., 2021). Thus, our evaluations are consistent with the previous studies with a reasonable variation range. Additionally, previous studies demonstrated that the performance of different SM products varies from one site to the other (Xing et al., 2021; Bi et al., 2016; Zeng et al., 2015). Generally, the SM products overestimate SM under dry condition, while underestimate SM under wet condition, and the SM products show worse performance in the sites with strong landscape heterogeneity (Zeng et al., 2015; Qu et al., 2019; Xing et al., 2021). Our study showed similar results that the bias is negatively correlated with the soil water content significantly for the three SM products (Figure R3). In summary, the evaluations of SM products based on our SM dataset are consistent with the previous studies. The discussion about the quality of SM datasets has been revised in the manuscript.

Table R1 the comparison of the evaluation metrics for ERA5_Land, GLDAS_Noah, and SMAP_L4 from previous studies over QTP with that of our study (mean value)

| product | layer | bias | | ubRMSE | | R | | source |
|---|---|---|---|---|---|---|---|---|
| | | reference | our study | reference | our study | reference | our study | |
| ERA5 _Land | surface | -0.01~0.59 | 0.179 | 0.018~0.095 | 0.032 | -0.23~0.84 | 0.471 | (Xing et al., 2021) |
| | profile | 0.04~0.45 | 0.190 | 0.013~0.089 | 0.024 | -0.22~0.96 | 0.323 | |
| GLDAS _Noah | surface | -0.15~0.108 | 0.023 | 0.037~0.058 | 0.032 | 0.287~0.785 | 0.296 | (Bi et al., 2016; Qu et al., 2019; Xing et al., 2021) |
| | subsurface | -- | 0.013 | 0.021~0.054 | 0.023 | -0.245~0.824 | 0.127 | |
| | profile | -0.24~0.16 | 0.014 | 0.005~0.084 | 0.022 | -0.19~0.79 | 0.164 | |
| SMAP _L4 | surface | -0.20~0.16 | -0.011 | 0.017~0.086 | 0.030 | 0.14~0.78 | 0.490 | (Xing et al., 2021) |
| | profile | -0.16~0.14 | 0.045 | 0.01~0.08 | 0.021 | 0.24~0.94 | 0.371 | |

[Figure]

Figure R3. The relationship between the bias of different SM products and average SM at surface layer.

☐ The evaluations of SM products are consistent with the previous results over QTP (Xing et al., 2021; Bi et al., 2016; Qu et al., 2019).

For the SHP dataset, Zhao et al. (2018) evaluated the soil properties products based on measurements at Maqu, Naqu, and Ngari stations over QTP. The comparisons of our evaluation and results of Zhao et al. (2018) are shown in Table R2. Zhao et al. (2018) found that the evaluation of SHP datasets varied significantly at different regions of QTP. Table R2 shows that our evaluation is consistent with the results of Maqu site, which is the nearest site to our study area.

Table R2. The comparisons of the bias of the evaluation of SHP datasets (HWSD and Soilgrid_250m) between Zhao et al. (2018) and this study.

| Dataset | site | clay (%) | silt (%) | sand (%) | BD (g/cm$^3$) | Source |
|---------|------|----------|----------|----------|---------------|--------|
| HWSD | Ngari | 12.60 | 21.30 | -34.20 | -0.17 | (Zhao et al., 2018) |
| | Naqu | 10.90 | 10.50 | -21.50 | -0.19 | |
| | Maqu | 6.49 | -28.00 | 21.90 | 0.36 | |
| | Heihe | 13.16 | -22.38 | 9.43 | 0.21 | This study |
| Soilgrid250 | Ngari | 8.43 | 14.70 | -22.80 | -0.27 | (Zhao et al., 2018) |
| | Naqu | 9.33 | 13.20 | -22.50 | -0.37 | |
| | Maqu | 9.70 | -20.50 | 11.40 | 0.12 | |
| | Heihe | 11.48 | -24.15 | 12.78 | 0.02 | This study |

Furthermore, the values of measured soil properties were compared to those available in the literature to cross-check whether they are within a reasonable range. Firstly, we found that the SHPs varied significantly at different sites. For example, the sand content is $84.54 \pm 8.28$ (mean $\pm$ standard deviation) and $26.95 \pm 10.55$ for the Ngari and Maqu, respectively (Zhao et al., 2018). Thus, we only compare our results to the previous studies near our study area (at Qilian Mountain, Table R3). Table R3 shows that the values of SHPs in our study have a reasonable range.

Table R3 the comparisons of SHP values in our dataset (mean±standard deviation) with the value ranges of previous studies in the Qilian Mountains

| SHP | Value | Source |
|---|---|---|
| clay | 6.66±1.49 | this study |
| (%) | 0.3~19 | (Hu et al., 2019; Hu et al., 2020; Yang et al., 2017; Zhi et al., 2017) |
| silt | 64.1±12.31 | this study |
| (%) | 17~90 | (Hu et al., 2016; Hu et al., 2019; Hu et al., 2020; Yang et al., 2017; Zhi et al., 2017) |
| sand | 29.23±13.25 | this study |
| (%) | 1~79 | (Hu et al., 2016; Hu et al., 2019; Hu et al., 2020; Yang et al., 2017; Zhi et al., 2017) |
| SOC | 4.53±4.03 | this study |
| (%) | 0.07~27 | (Hu et al., 2016; Hu et al., 2019; Song et al., 2016; Zhi et al., 2017) |
| BD | 1.17±0.24 | this study |
| (g/cm$^3$) | 0.3~1.8 | (Hu et al., 2019; Hu et al., 2020; Yang et al., 2017; Yang et al., 2020; Zhi et al., 2017) |
| $\log_{10}K_S$ | 1.4±0.5 | this study |
| (cm/day) | 0.07~3.6 | (Hu et al., 2020; Liu et al., 2013; Yang et al., 2020) |

Moreover, collecting the SM and SHP datasets since 2013, we have done lots of work based on the datasets. For example, based on the SHP dataset, we have analyzed the spatial distribution of the saturated soil hydraulic conductivity and soil properties (Zhao et al., 2014; Tian et al., 2017), exploring the scaling method of soil texture (Li et al., 2018), the application and improvement of hydrological models (Jin et al., 2015; Zhang et al., 2016). Based on the SM dataset, we have done the analysis of the spatial-temporal variation of SM (Tian et al., 2019), estimation of soil water storage (Tian et al., 2020), analysis of preferential flow (Kang et al., 2022), validation and improvement of the SM products (Zhang et al., 2017; 2019; Bai et al., 2020), and the validation of hydrological models (Su et al., 2020).

Lastly, the saturated soil hydraulic conductivity ($K_S$, Tian et al., 2017) of our dataset has contributed

as a representative $K_S$ network in the SoilKsatDB, a global database of $K_S$ (Gupta et al., 2021, Earth System Science Data).

In summary, the comparisons of our results with the previous studies, cross-checking with the available literature, and the applications of our SHP and SM datasets indicate that the quality of our dataset is good and has been accepted by research community. The discussion about the quality of the datasets has been revised in the manuscript.

☐ Our evaluations of SHP datasets are consistent with the results of Zhao et al. (2018).

☐ Additionally, the value ranges of SHPs in our dataset are consistent with the previous studies within the study area (Hu et al., 2016; Hu et al., 2019; Hu et al., 2020; Liu et al., 2013; Song et al., 2016; Yang et al., 2017; Yang et al., 2020; Zhi et al., 2017).

**RC: I believe that the long-time series point SM measurements are valuable and meaningful. However, I do not think they are suitable for validating coarse SM products. Since the in-situ measurements are all obtained from single stations, spatial heterogeneity impacts on the validation results can not be ignored. They should be considered, especially when evaluating SM products with a spatial resolution of tens kilometers using in-situ point measurements. Actually, dense in-situ SM observation networks are an effective way to minimize the impacts of spatial heterogeneity. Several dense in-situ SM observation networks within the Qinghai Tibet Plateau, such as Heihe network constructed during HiWATER, Naqu and Pali of the CTP-SMTMN networks and Maqu and Ngari of the Tibet-Obs networks, have provided long time-series SM measurements which can be well used for SM evaluation. Therefore, I would suggest the authors use point SM measurements in different applications on a small scale to clarify their quality.**

AR: Thanks for your suggestion. On the one hand, the mismatch of scale between the SM measurements

and the SM products is still a challenge in validating SM products (Jin et al., 2017), particularly in hard to reach, data lacking, topographically complex high mountain areas such as the Qilian Mountain Ranges and QTP, and has been discussed in the revised manuscript. On the other hand, the point SM measurements have been applied successfully as follows: (1) The response of SM at different layers to rainfall under different land covers and its control factors (Tian et al., 2019, Agricultural and Forest Meteorology, 271(15): 225-239). (2) The coupling of surface SM and subsurface SM and the estimation of profile SM from surface SM (Tian et al 2020, Hydrology and Earth System Sciences, 24(9): 4659-4674). (3) Analysis of the occurrence and controls of preferential flow (Kang et al., 2022, Journal of Hydrology, 607: 127528). (4) The estimation of rainfall from SM (Lai et al., 2022, Journal of Hydrology, 606: 127430). (5) Validation of hydrological modeling at different sites (Su et al., 2020, Journal of Geophysical Research: Atmospheres, 125(18): e2020JD032727). (6) Evaluation of the SMAP and SMOS products using our SM dataset (Zhang et al., 2017, Remote Sensing, 9(11): 1111; 2019, Science China Earth Sciences, 62(4): 703-718).

Additionally, based on the point SM measurements, we also analyzed the variation of SM with elevation in this study, and we find the increasing of SM with the increasing elevation (Figure S4 in the supplement), which is consistent with the previous studies (Geng et al., 2017).

[Figure]

Figure S4. The variation of SM with elevation for different soil depths based on the observed temporal

mean value (and standard deviation) at each station and the station elevation, including the linear fitting result with the 95% confidence band. L1 represents SM for layer 1, ** indicates the slope of the linear regression is significant at the 0.01 level.

☐ Notably, the scale mismatch between the point location of in-situ SM measurements and the footprints of SM products will introduce additional errors in the evaluation of the SM products (Jin et al., 2017).

**Specific Comments**

**RC: L40, it is arbitrary to say "highly uncertain".**

AR: We have changed the statement in the revised manuscript.

☐ the uncertainty associated with information about SHP datasets and SM products still exists

**RC: L105, in our stud area**

AR: We have changed it to "in our study area".

**RC: L130, make sure that it is "at the long-term SM monitoring stations" or "at the random sampling**

AR: We have changed it to "from each layer of long-term SM monitoring station".

**RC: Table 1, suggest to list spatial resolutions for HWSD, SoilGrid, ShangYG and DaiYJ.**

AR: We have listed the spatial resolutions of SHP datasets in Table 1.

☐ Table 1. Calculation of SHPs at different depths (5 cm, 25 cm, 0–30 cm) for evaluating the soil property datasets.

| Depth | HWSD | SoilGrid | ZhangYG | DaiYJ | observation |
|---|---|---|---|---|---|
| Spatial resolution | 30″ | 250 m | 1 km | 30″ | - |
| 5 cm | - | $SG_{0-5}$ | $ZhangYG_{0-5}$ | $DaiYJ_{0-5}$ | $obs_5$ |
| 25 cm | - | $SG_{15-30}$ | - | - | $obs_{25}$ |
| 0-30 cm | $HWSD_{0-30}$ | $(5 \cdot SG_{0-5}+10 \cdot SG_{5-15}+15 \cdot SG_{15-30})/30$ | - | - | $(obs_5+obs_{25})/2$ |

Note: *SG* and *obs* represent soil properties from the SoilGrid dataset and from observations, respectively. The subscript gives the soil depths for the SoilGrids properties, and "-" indicates that data for a specific layer are not available in the datasets.

**RC: Table 2, suggest to list spatial resolutions for GLDAS, ERA5 and SMAP SM products.**

AR: We have listed the spatial resolutions of SM products in Table 2.

☐ Table 2. Calculation of the soil moisture at different depths (surface, subsurface and profile) from different datasets.

| Product | Spatial resolution | Surface (0-10 cm) | Subsurface (10-100 cm) | Profile (0-100 cm) |
|---|---|---|---|---|
| GLDAS | 0.25° | $sm_{0-10}$ | $(30 \cdot sm_{10-40}+60 \cdot sm_{40-100})/90$ | $(10 \cdot sm_{0-10}+30 \cdot sm_{10-40}+60 \cdot sm_{40-100})/100$ |
| ERA5_Land | 9 km | $(7 \cdot sm_{0-7}+3 \cdot sm_{7-28})/10$ | $(18 \cdot sm_{7-28}+72 \cdot sm_{28-100})/90$ | $(7 \cdot sm_{0-7}+21 \cdot sm_{7-28}+72 \cdot sm_{28-100})/100$ |
| SMAP_L4 | 9 km | $sm_{0-10}$ | $(10 \cdot sm_{0-100}-sm_{0-10})/9$ | $sm_{0-100}$ |
| Observation | - | $sm_5$ | $(10 \cdot sm_{15}+10 \cdot sm_{25}+20 \cdot sm_{40}+50 \cdot sm_{60})/90$ | $(10 \cdot sm_5+10 \cdot sm_{15}+10 \cdot sm_{25}+20 \cdot sm_{40}+50 \cdot sm_{60})/100$ |

Note: *sm* is soil moisture, and subscripts indicate the range of depth. For the derived products, the depth is a range (e.g., 0-10, representing 0–10 cm), while for the observations, *sm* is reported at the observed depths of 5 cm, 15 cm, 25 cm, 40 cm, and 60 cm.

**RC: Line 230, double-check the numbers here. I can not well relate some of the numbers here to those listed in Table 3. For example, why n ranges within 0.09 and 0.12? why cv of clay ranges within (0.18, 0.28)? etc.**

AR: Thank you for your reminder, we have checked and revised the numbers thoroughly according to Table 3.

☐ The results show that $C_V$ of the *n* (ranged within 0.1–0.134 at different depths) of Van Genuchten model is less than 0.16, which is a relatively low spatial variability. Meanwhile, $C_V$ for clay ranged from 0.156 to 0.287, and ranged from 0.174 to 0.237 for silt. $C_V$ of $\theta_s$ ranged from 0.187 to 0.223, and the $C_V$ of BD ranged from 0.195 to 0.238. These SHPs therefore show a relatively moderate variability according to Wilding (1985). $C_V$ of sand ranged between 0.439 and 0.618, $\theta_r$ ranged

between 0.472 and 0.654, $C_V$ of $K_S$ ranged between 0.293 and 0.534, and $C_V$ of $\alpha$ ranged from 0.707 to 1.094. The $C_V$ values of these SHPs were mostly higher than 0.36, indicating strong spatial variability. Thus, the soil texture for clay and silt had the lowest spatial heterogeneity, while both $K_S$, $\theta_r$, and $\alpha$ had high spatial variability in the study area.

**Table 3. The descriptive statistics (median and coefficient of variation) for soil properties at different depths.**

| depth (cm) | clay (%) | silt (%) | sand (%) | BD* (g/cm³) | SOC (%) | $\log_{10}K_S$* (cm/d) | $\alpha$ (cm⁻¹) | $n$ (-) | $\theta_s$* (cm³/cm³) | $\theta_r$ (cm³/cm³) |
|---|---|---|---|---|---|---|---|---|---|---|
| 5 | 6.600 (0.224) | 66.562 (0.192) | 26.686 (0.453) | 1.153 (0.207) | 4.024 (0.889) | 1.445 (0.356) | 0.022 (0.896) | 1.370 (0.119) | 0.550 (0.189) | 0.107 (0.564) |
| 15 | 6.356 (0.156) | 66.607 (0.174) | 26.008 (0.439) | 1.151 (0.195) | 1.466 (1.019) | 1.564 (0.293) | 0.026 (0.877) | 1.382 (0.100) | 0.533 (0.187) | 0.126 (0.472) |
| 25 | 6.381 (0.243) | 65.776 (0.237) | 27.362 (0.513) | 1.189 (0.228) | 2.077 (0.945) | 1.454 (0.381) | 0.026 (0.707) | 1.349 (0.134) | 0.514 (0.214) | 0.094 (0.654) |
| 40 | 6.656 (0.203) | 64.831 (0.222) | 29.213 (0.505) | 1.189 (0.233) | 1.327 (0.998) | 1.436 (0.405) | 0.027 (0.948) | 1.412 (0.117) | 0.462 (0.223) | 0.129 (0.603) |
| 60 | 6.848 (0.287) | 70.089 (0.233) | 20.724 (0.618) | 1.238 (0.238) | 1.240 (1.05) | 1.114 (0.534) | 0.021 (1.094) | 1.394 (0.104) | 0.502 (0.216) | 0.146 (0.544) |
| all | 6.552 (0.227) | 66.323 (0.21) | 26.85 (0.486) | 1.168 (0.219) | 1.473 (1.008) | 1.449 (0.37) | 0.024 (0.887) | 1.374 (0.117) | 0.526 (0.200) | 0.113 (0.566) |

*indicates that the soil property is significantly (p < 0.05) different at different depths. $\log_{10}K_S$ is the $\log_{10}$ transformed $K_S$. $\alpha$ and $n$ are the parameter of Van Genuchten model (Supplement). $\theta_s$ and $\theta_r$ are the saturated SM and residual SM, respectively (Supplement).

**RC: L250, is it -0.66?**

AR: Sorry for the error. It has been changed to "-0.66" in the revised manuscript.

**RC: L255, wrong space place in "… significant ,except…"**

AR: Thank you. It has been changed to "significant, except"

**RC: Figure 5, keep the soil property name consistent with those in the text. E.g., bulk to bulk density or BD, s in logks should be a subscript**

AR: Thank you, the names of SHPs have been changed to be consistent with those in the text (Figure 5).

[Figure]

Figure 5. Correlations between different SHPs. Different colors indicate different soil layers. The lower triangle of the figure area shows the scatterplots between different SHPs of different layers. The upper triangle lists the Peason's correlation coefficients (R) of different SHPs, and the first number in each box (Corr: ) is the R value calculated by combining data from all soil depths. Plots running diagonally across the figure area represent the distribution of each SHP.

**RC: L280, Theta_r and theta_s should be written formally.**

AR: We have changed the Theta_r and Theta_s into $\theta_r$ and $\theta_s$, respectively (Figure 6).

[Figure]

Figure 6. The spatial distribution of soil texture (sand, silt, clay, %), BD (g/cm$^3$), $\log_{10}K_S$ ($\log_{10}$ transformed $K_S$, cm/d), $\theta_r$ (cm$^3$/cm$^3$), $\theta_s$ (cm$^3$/cm$^3$), $\alpha$, and $n$ in the study area.

**RC: Figure 7, bulk, theta_s, Theta_r, and alpha should be written formally.**

AR: We have changed the Theta_r, Theta_s, and bulk into $\theta_r$, $\theta_s$, and BD, respectively (Figure 7).

[Figure]

Figure 7. (a) The vertical distribution of sand, clay, silt, $\log_{10}K_S$, BD, SOC and the parameters of the Van Genuchten model for fitting the soil water retention curve ($\theta_r$, $\theta_s$, $\alpha$, and $n$) in the study area; (b) boxplots show the distributions of the soil water retention curves for different soil layers.

**RC:** **L330, Figure 11 here should Figure 10? The "higher" in "…and this is higher than the temporal…" should be "lower"?**

AR: Thank you. Yes, it should be Figure 10. And the $C_V$ of temporal variation should be lower than that of spatial variation (Figure 10). We have changed it in the revised manuscript.

☐    The spatial variability of SM varies from 0.32 to 0.65 over the study area, with a mean $C_V$ of 0.45, and this is higher than the temporal variability of SM over the study period (2014–2020), which varied from 0.03 to 0.52, with a mean $C_V$ of 0.18 (Figure 10).

[Figure]

Figure 10. (a) The relationship between the spatial $C_V$ and mean SM for each month for different soil layers; (b) the relationship between the temporal $C_V$ and mean SM for each station for different soil layers. The curve and shading in each plot show the fitted curve and the 95% confidence interval, and the legend shows the fitting equation between $C_V$ and mean SM for each soil layer. [**] shows where the fitted curves are significant at the 0.01 level.

**RC: Figure 9, the text 0.75 in the legend is wrong? Should it be 0.075? In addition, clarify in the text how to obtain figure 9? Same as figure 6 using Kriging method in ArcGIS to interpolate?**

AR: Thank you, the value should be 0.075 and the spatial distribution of SM is obtained through the Kriging method in ArcGIS. We have changed the value and clarified the method in the revised manuscript (Figure 9).

[Figure]

Figure 9. (a)-(e) The spatial distribution of the average SM during the study period for (a) layer 1 to (e) layer 5 through the Kriging method. Circles with different sizes show SM measurements from stations with different mean values over the study period. (f) The variation of the average SM with depth in different years, the line and shading show the fitted curve and the 95% confidence interval.

**RC: L350, two meanings for PBIAS here. One is positive bias, the other is percent bias.**

AR: Yes. The PBIAS can reflect both the positive or negative bias of the datasets and the percent of the bias. As the value ranges of SHPs such as soil texture and bulk density varied at a different order of magnitude, we used PBIAS instead of "bias" to compare the evaluations of different SHP datasets. The equation of the metrics used in this study has been listed in the revised supplement.

$$PBIAS = 100 \times \frac{\sum(M-O)}{\sum O} \tag{1}$$

where *M, O* are the estimation values and observations, respectively.

**RC: Figure 11, use BD instead of bulk.**

AR: We have changed it in Figure 11.

[Figure]

Figure 11. (a) Boxplots show the distributions for the evaluated derived datasets (soil texture and BD from the SoilGrid and HWSD datasets, $\log_{10}K_S$ from the ZhangYG and DaiYJ datasets), with the corresponding distributions for the observations. (b) Metrics to compare the existing derived soil datasets with the corresponding observations in the study area.

**RC: L410, how can you conclude that "our SM dataset provides new accurate in-situ SM measurements covering …"?**

AR: We have changed our statement in the revised manuscript.

☐  Our SM dataset provides an in-situ SM measurements covering 2014–2020 over a large-scale mountainous area

References:

Bai, X., Zhang, L., He, C., and Zhu, Y.: Estimating Regional Soil Moisture Distribution Based on NDVI and Land Surface Temperature Time Series Data in the Upstream of the Heihe River Watershed, Northwest China, Remote Sens., 12, 2414, https://doi.org/10.3390/rs12152414, 2020. (by our team members in the Heihe River Watershed in the Qilian Mountain Ranges)

Bi, H., Ma, J., Zheng, W., and Zeng, J.: Comparison of soil moisture in GLDAS model simulations and in situ observations over the Tibetan Plateau, J. Geophys. Res. Atmos., 121, 2658-2678,

https://doi.org/10.1002/2015JD024131, 2016.

Chan, S. K., Bindlish, R., O'Neill, P. E., Njoku, E., Jackson, T., Colliander, A., Chen, F., Burgin, M., Dunbar, S., and Piepmeier, J.: Assessment of the SMAP passive soil moisture product, IEEE Trans. Geosci. Remote Sens., 54, 4994-5007, https://doi.org/10.1109/TGRS.2016.2561938, 2016.

Cobos, D. R., and Chambers, C.: Calibrating ECH2O soil moisture sensors, Application Note, Decagon Devices, Pullman, Washington, 2010.

Geng, H., Pan, B., Huang, B., Cao, B., and Gao, H.: The spatial distribution of precipitation and topography in the Qilian Shan Mountains, northeastern Tibetan Plateau, Geomorphology, 297, 43-54, https://doi.org/10.1016/j.geomorph.2017.08.050, 2017.

Guo, L., Lin, H., Fan, B., Nyquist, J., Toran, L., and Mount, G. J.: Preferential flow through shallow fractured bedrock and a 3D fill-and-spill model of hillslope subsurface hydrology, J. Hydrol., 576, 430-442, https://doi.org/10.1016/j.jhydrol.2019.06.070, 2019.

Guo, L., Mount, G. J., Hudson, S., Lin, H., and Levia, D.: Pairing geophysical techniques improves understanding of the near-surface Critical Zone: Visualization of preferential routing of stemflow along coarse roots, Geoderma, 357, https://doi.org/10.1016/j.geoderma.2019.113953, 2020.

Gupta, S., Hengl, T., Lehmann, P., Bonetti, S., and Or, D.: SoilKsatDB: global database of soil saturated hydraulic conductivity measurements for geoscience applications, Earth Syst. Sci. Data, 13, 1593-1612, https://doi.org/10.5194/essd-13-1593-2021, 2021.

Hu, G.R., Li, X.Y., and Yang, X.F.: The impact of micro-topography on the interplay of critical zone architecture and hydrological processes at the hillslope scale: Integrated geophysical and hydrological experiments on the Qinghai-Tibet Plateau, J. Hydrol., 583, 124618, https://doi.org/10.1016/j.jhydrol.2020.124618, 2020.

Hu, J., Lü, D., Sun, F., Lü, Y., Chen, Y., and Zhou, Q.: Soil Hydrothermal Characteristics among Three Typical Vegetation Types: An Eco-Hydrological Analysis in the Qilian Mountains, China, Water, 11, 1277, https://doi.org/10.3390/w11061277, 2019.

Hu, X., Li, Z.-C., Li, X.-Y., and Liu, L.-y.: Quantification of soil macropores under alpine vegetation using computed tomography in the Qinghai Lake Watershed, NE Qinghai–Tibet Plateau, Geoderma, 264, 244-251, https://doi.org/10.1016/j.geoderma.2015.11.001, 2016.

Jin, R., Li, X., and Liu, S. M.: Understanding the Heterogeneity of Soil Moisture and Evapotranspiration Using Multiscale Observations From Satellites, Airborne Sensors, and a Ground-Based Observation Matrix, IEEE Geosci. Remote Sens. Lett., 14, 2132-2136, https://doi.org/10.1109/LGRS.2017.2754961, 2017.

Jin, X., Zhang, L. h., Gu, J., Zhao, C., Tian, J., and He, C. S.: Modeling the impacts of spatial heterogeneity in soil hydraulic properties on hydrological process in the upper reach of the Heihe River in the Qilian Mountains, Northwest China, Hydrol. Processes, 29, 3318-3327, https://doi.org/10.1002/hyp.10437, 2015. (by our team members in the Heihe River Watershed in the Qilian Mountain Ranges)

Kang, W., Tian, J., Lai, Y., Xu, S., Gao, C., Hong, W., Zhou, Y., Pei, L., and He, C.: Occurrence and controls of preferential flow in the upper stream of the Heihe River Basin, Northwest China, J. Hydrol., 607, 127528, https://doi.org/10.1016/j.jhydrol.2022.127528, 2022. (by our team members in the Heihe River Watershed in the Qilian Mountain Ranges)

Lai, Y., Tian, J., Kang, W., Gao, C., Hong, W., and He, C.: Rainfall estimation from surface soil moisture using SM2RAIN in cold mountainous areas, J. Hydrol., 127430, https://doi.org/10.1016/j.jhydrol.2022.127430, 2022. (by our team members in the Heihe River Watershed in the Qilian Mountain Ranges)

Li, J., Zhang, L., He, C., and Zhao, C.: A Comparison of Markov Chain Random Field and Ordinary Kriging Methods for Calculating Soil Texture in a Mountainous Watershed, Northwest China, Sustainability, 10, 2819, https://doi.org/10.3390/su10082819, 2018. (by our team members in the Heihe River Watershed in the Qilian Mountain Ranges)

Liang, W. L., Kosugi, K. I., and Mizuyama, T.: Soil water dynamics around a tree on a hillslope with or without rainwater supplied by stemflow, Water Resour. Res., 47, 2144-2150, https://doi.org/10.1029/2010WR009856, 2011.

Liu, H., Zhao, W. Z., and He, Z. B.: Self-organized vegetation patterning effects on surface soil hydraulic conductivity: A case study in the Qilian Mountains, China, Geoderma, 192, 362-367, https://doi.org/10.1016/j.geoderma.2012.08.008, 2013.

Qu, Y., Zhu, Z., Chai, L., Liu, S., Montzka, C., Liu, J., Yang, X., Lu, Z., Jin, R., Li, X., Guo, Z., and Zheng, J.: Rebuilding a Microwave Soil Moisture Product Using Random Forest Adopting AMSR-E/AMSR2 Brightness Temperature and SMAP over the Qinghai–Tibet Plateau, China, Remote Sens., 11, 683, https://doi.org/10.3390/rs11060683, 2019.

Song, X. D., Brus, D. J., Liu, F., Li, D.-C., Zhao, Y.-G., Yang, J.-L., and Zhang, G.-L.: Mapping soil organic carbon content by geographically weighted regression: A case study in the Heihe River Basin, China, Geoderma, 261, 11-22, https://doi.org/10.1016/j.geoderma.2015.06.024, 2016.

Su, T., Zhang, B., He, X., Shao, R., Li, Y., Tian, J., Long, B., and He, C.: Rational Planning of Land Use Can Maintain Water Yield Without Damaging Ecological Stability in Upstream of Inland River: Case Study in the Hei River Basin of China, J. Geophys. Res. Atmos., 125, e2020JD032727, https://doi.org/10.1029/2020JD032727, 2020. (by our team members in the Heihe River Watershed in the Qilian Mountain Ranges)

Tian, J., Zhang, B. Q., He, C. S., and Yang, L. X.: Variability in Soil Hydraulic Conductivity and Soil Hydrological Response Under Different Land Covers in the Mountainous Area of the Heihe River Watershed, Northwest China, Land Degrad. Dev., 28, 1437-1449, https://doi.org/10.1002/ldr.2665, 2017. (by our team members in the Heihe River Watershed in the Qilian Mountain Ranges)

Tian, J., Zhang, B. Q., He, C. S., Han, Z. B., Bogena, H. R., and Huisman, J. A.: Dynamic response patterns of profile soil moisture wetting events under different land covers in the Mountainous area of the Heihe River Watershed, Northwest China, Agric. For. Meteorol., 271, 225-239, https://doi.org/10.1016/j.agrformet.2019.03.006, 2019. (by our team members in the Heihe River Watershed in the Qilian Mountain Ranges)

Tian, J., Han, Z., Bogena, H. R., Huisman, J. A., Montzka, C., Zhang, B., and He, C.: Estimation of subsurface soil moisture from surface soil moisture in cold mountainous areas, Hydrol. Earth Syst. Sci., 24, 4659-4674, https://doi.org/10.5194/hess-24-4659-2020, 2020. (by our team members in the Heihe River Watershed in the Qilian Mountain Ranges)

Wilding, L. P.: Spatial variability: Its documentation, accommodation and implication to soil survey, Spatial Variations, 166-194 pp., 1985.

Xing, Z., Fan, L., Zhao, L., De Lannoy, G., Frappart, F., Peng, J., Li, X., Zeng, J., Al-Yaari, A., Yang, K., Zhao, T., Shi, J., Wang, M., Liu, X., Hu, G., Xiao, Y., Du, E., Li, R., Qiao, Y., Shi, J., Wen, J., Ma, M., and Wigneron, J.-P.: A first assessment of satellite and reanalysis estimates of surface and root-zone soil moisture over the permafrost region of Qinghai-Tibet Plateau, Remote Sens. Environ., 265, 112666, https://doi.org/10.1016/j.rse.2021.112666, 2021.

Yang, J. J., He, Z. B., Du, J., Chen, L. F., Zhu, X., Lin, P. F., and Li, J.: Soil water variability as a function of precipitation, temperature, and vegetation: a case study in the semiarid mountain region of China, Environ. Earth Sci., 76, 206, https://doi.org/10.1007/s12665-017-6521-0, 2017.

Yang, Y., Chen, R.S., Song, Y.X., Han, C.T., Liu, Z.W., and Liu, J.F.: Spatial variability of soil hydraulic conductivity and runoff generation types in a small mountainous catchment, Journal of Mountain Science, 17, 2724-2741, https://doi.org/10.1007/s11629-020-6258-1, 2020.

Zeng, J., Li, Z., Chen, Q., Bi, H., Qiu, J., and Zou, P.: Evaluation of remotely sensed and reanalysis soil moisture products over the Tibetan Plateau using in-situ observations, Remote Sens. Environ., 163, 91-110, https://doi.org/10.1016/j.rse.2015.03.008, 2015.

Zhang, L., Jin, X., He, C., Zhang, B., Zhang, X., Li, J., Zhao, C., Tian, J., and Demarchi, C.: Comparison of SWAT and DLBRM for Hydrological Modeling of a Mountainous Watershed in Arid Northwest China, J. Hydrol. Eng., 21, https://doi.org/10.1061/(ASCE)HE.1943-5584.0001313, 2016. (by our team members in the Heihe River Watershed in the Qilian Mountain Ranges)

Zhang, L., He, C., and Zhang, M.: Multi-Scale Evaluation of the SMAP Product Using Sparse In-Situ

Network over a High Mountainous Watershed, Northwest China, Remote Sens., 9, 1111, https://doi.org/10.3390/rs9111111, 2017. (by our team members in the Heihe River Watershed in the Qilian Mountain Ranges)

Zhang, L., He, C., Zhang, M., and Zhu, Y.: Evaluation of the SMOS and SMAP soil moisture products under different vegetation types against two sparse in situ networks over arid mountainous watersheds, Northwest China, Sci. China Earth Sci., 62, 703-718, https://doi.org/10.1007/s11430-018-9308-9, 2019. (by our team members in the Heihe River Watershed in the Qilian Mountain Ranges)

Zhang, Y., Zhao, W., He, J., and Fu, L.: Soil Susceptibility to Macropore Flow Across a Desert-Oasis Ecotone of the Hexi Corridor, Northwest China, Water Resour. Res., 54, https://doi.org/10.1002/2017WR021462, 2018.

Zhao, C., Zhang, L., Li, J., Tian, J., Wu, W., Jin, X., Zhang, X., Jiang, Y., Wwang, X., He, C., and Bai, X.: Analysis of the relationships between the spatial variations of soil moisture and the environmental factors in the upstream of the Heihe River watershed, J. Lanzhou Univ. Nat. Sci., 50, 338-347, https://doi.org/10.13885/j.issn.0455-2059.2014.03.008, 2014. (in Chinese) (by our team members in the Heihe River Watershed in the Qilian Mountain Ranges)

Zhao, H., Zeng, Y., Lv, S., and Su, Z.: Analysis of soil hydraulic and thermal properties for land surface modeling over the Tibetan Plateau, Earth Syst. Sci. Data, 10, 1031-1061, https://doi.org/10.5194/essd-10-1031-2018, 2018.

Zhi, J., Zhang, G., Yang, F., Yang, R., Liu, F., Song, X., Zhao, Y., and Li, D.: Predicting mattic epipedons in the northeastern Qinghai-Tibetan Plateau using Random Forest, Geoderma Regional, 10, 1-10, https://doi.org/10.1016/j.geodrs.2017.02.001, 2017.

---

## Author Comment (AC3)

**RC: Reviewer Comment**,   AR: Author Response,   ☐ New Manuscript text

Dear Referee,

We would like to thank you very much for your effort in reviewing our manuscript. Please find our responses to your comments below. These should be considered as preliminary (part of the interactive discussion) as implementation of the final changes also depends on another referee report that is still pending.

Kind regards,

Dr. Prof. Chansheng He

(on behalf of all the authors)

**The comments of editor and reviewers are in black with bold text,** the author's answers are indicated in blue color, as well as old text passages. New text passages are indicated in green color.

**General Comments**

**This paper provides potentially a very useful and important dataset. Substantial effort to collect soil samples and build up a long-term SM monitoring network in the high and cold mountainous region. Potentially a good candidate for ESSD. However, the important first-hand measured data cannot be accessible, for instance, SMST at the half-hourly scale on 32 LULC-Soil-DEM zones and measured SWRCs and possible soil heat conductivity curves, which hampers its potential to become a useful dataset in the hydrology, RS and soil research conducted on the high and cold mountainous region. The reviewer suggests the author uploading all raw data and completing the description data. Moreover, provide a brief description of the loaded data (in the data availability) that is consistent with the description in the manuscript. For detailed comments please see below. Some comments are labeled in the .pdf.**

AR: We appreciate your comments and suggestion. Currently three of Ph.D. candidates are using the

collected soil moisture datasets to work on their Ph.D. theses and continue to expand the datasets as well. By international protocol, we cannot upload the raw SM datasets at this point of time. We are willing to share all the raw SM datasets later once the Ph.D. candidates complete their theses. Thank you for your understanding.

Meanwhile, as suggested by reviewer 1, the raw SHP datasets (including the soil texture, soil dry bulk density, soil organic carbon, soil saturated hydraulic conductivity and soil water retention curve) of eight representative stations (with five layers for each station) have been updated in the datasets of the manuscript (https://doi.org/10.5281/zenodo.6514191). The description of the dataset has also been uploaded to the public datasets of the manuscript.

**Specific Comments**

RC: **In Line 69 about 'a long-term SM dataset for the Qilian Mountains', the reviewer knows the focus of this dataset is more about SM, while soil temperature information measured by ECH2O 5TE device should also be released for a comprehensive soil physical property information, which is more helpful in the use of soil water and heat transport (in LSM) research conducted on the high and cold mountainous region, as well as microwave signal simulation and the corresponding SM retrieval validation.**

AR: Thanks for your suggestions. As stated by the reviewer, SM and soil temperature (ST) datasets are very useful for investigating the soil water and heat transport (in LSM), as well as microwave signal simulation and the corresponding SM retrieval validation. Our team is focusing on the land surface water-energy transport, runoff process, and land-atmosphere interactions in the Qilian Mountains through the regional climate model (e.g. Weather Research and Forecasting Model), land surface model (e.g. Community Land Model and NOAH-MultiParameterization Land Surface Model). In addition, we are also combining methods such as machine learning and data assimilation with remote sensing observations to produce soil moisture and temperature datasets with a high spatial and temporal resolution over the Qilian Mountains. The above work has not yet been completed. We are

therefore sorry that we are unable to update this dataset at this point of time but will upload the updated dataset as soon as the work is done.

**RC: In line 114-115, it is mentioned that SM at different soil depths with a time interval of 30 min. The reviewer does notice this half-hourly data cannot be accessible. The reviewer suggests publishing SMST at the measured time scale rather than at the processed scale. Moreover, the reviewer does not think evaluating SM at the monthly scale is a routine, at least at the daily scale is more convincible.**

AR: As stated above, three of Ph.D. candidates are using the collected SM/ST datasets at 30 min or daily scale to work on their Ph.D. theses and continue to expand the datasets as well. By international protocol, we cannot upload the raw SM/ST datasets with the 30 min and daily scale at this point of time.

Also suggested by reviewer 1, the validations of the SM products at daily scale was performed. Results are shown in Figure S6 (scatterplots comparing the different SM products with observations) and Figure S7 (the results of the evaluation metrics for the different products). Results indicate that the performance of different SM products at daily scale is consistent with that at monthly scale. This part has been added to the revised supplement.

[Figure]

Figure S6. Scatterplots comparing the different derived SM products with the observed SM for different soil layers at daily scale. The metrics within each plot show the median value of the metrics for all stations. The smoothed color density in the scatter plots shows the density of points.

[Figure]

Figure S7. Metrics for comparing the different SM products (GLDAS2.1_Noah, ERA5_Land, SMAP_L4) with the in-situ SM observations for different soil layers at daily scale. The different letters above the violin plot indicate the significant differences ($p < 0.05$) between different products for each soil layer.

**RC: L130, In the sheet ''station information' of the uploaded file 'soil moisture data_NE_QTP.xlsx', there is no information of land use/type data, elevation, soil type and soil texture. In line 123, 'Environmental factors such as the position, slope, aspect, root depth, and land cover were measured at each station', please complete all these related information. In Figure 1, please also add the meaning of 32 main LULC (Land Use/Cover)- soil-DEM types, which is not clear for the reviewer who does not concern with LULC research.**

AR: The above information (including the position, land use/type, elevation, soil, slope, and root information) has been uploaded to the public datasets of this manuscript. The meaning of the 32 LULC (Land Use/Cover)- soil-DEM types have also been added to the supplement (Table S1)

**Table S1. The meaning of the 32 LULC-soil-DEM types in the study area**

| ID | Elevation (m) | Soil Genetic Classification | Land use/cover | Percentage (%) [a] |
|----|---------------|------------------------------|----------------|---------------------|
| D1 | 2000-2500 | Typical sierozem | Middle coverage grassland | 1.07 |
| D2 | 2000-2500 | Light chestnut soil | Middle coverage grassland | 2.61 |
| D3 | 2000-2500 | Sliming grey desert soil | Middle coverage grassland | 2.43 |
| D4 | 2500-3000 | Typical chestnut soil | Middle coverage grassland | 0.84 |
| D5 | 2500-3000 | Light chestnut soil | Middle coverage grassland | 1.91 |
| D6 | 2500-3000 | Light chestnut soil | Middle coverage grassland | 2.53 |
| D7 | 3000-3500 | Calcareous alpine steppe soil | Middle coverage grassland | 2.74 |
| D8 | 3000-3500 | Saturation alpine steppe soil | Middle coverage grassland | 2.97 |
| D9 | 3500-4000 | Saturation alpine steppe soil | Middle coverage grassland | 9.44 |
| D10 | 3500-4000 | Calcareous alpine steppe soil | Middle coverage grassland | 9.56 |
| D11 | 2500-3000 | Typical chestnut soil | Forestland | 1.53 |
| D12 | 2500-3000 | Typical grey cinnamon soil | Forestland | 1.32 |
| D13 | 2500-3000 | Peat subalpine steppe soil | Forestland | 3.45 |
| D14 | 2500-3000 | Light chestnut soil | Forestland | 2.14 |
| D15 | 3000-3500 | Peat subalpine steppe soil | Forestland | 5.52 |
| D16 | 3000-3500 | Saturation alpine steppe soil | Forestland | 2.23 |
| D17 | 3500-4000 | Peat subalpine steppe soil | Forestland | 2.21 |
| D18 | 2500-3000 | Typical chernozem | Farmland [b] | 0.96 |
| D19 | 2500-3000 | Dry chernozem | Farmland [b] | 1.01 |
| D20 | 2500-3000 | Typical chestnut soil | Barren land | 1.29 |
| D21 | 2500-3000 | Calcareous alpine steppe soil | Barren land | 3.02 |
| D22 | 3000-3500 | Calcareous alpine steppe soil | Barren land | 8.77 |
| D23 | 3000-3500 | Saturation alpine steppe soil | Barren land | 0.77 |
| D24 | 3500-4000 | Typical alpine steppe soil | Barren land | 2.08 |
| D25 | 4000-4500 | Typical alpine frost desert soil | Barren land | 1.96 |
| D26 | 4000-4500 | Saturation alpine steppe soil | Barren land | 0.80 |
| D27 | 2500-3000 | Typical chestnut soil | High coverage grassland | 1.18 |
| D28 | 2500-3000 | Light chestnut soil | High coverage grassland | 0.78 |
| D29 | 3000-3500 | Typical chestnut soil | High coverage grassland | 0.68 |
| D30 | 3000-3500 | Peat subalpine steppe soil | High coverage grassland | 0.80 |
| D31 | 3000-3500 | Saturation alpine steppe soil | High coverage grassland | 1.31 |
| D32 | 3500-4000 | Saturation alpine steppe soil | High coverage grassland | 1.64 |

Note: [a] Percentage (%) means the percentage of area of the DEM-Soil-LULC type in the study area; [b] For the Farmland: as the influence of agricultural activities, it's difficulty to monitor SM at the farmland for long term, we install the sensor in the field ridge of the farmland, and the actual land cover of the farmland site is high coverage grassland.

**RC: In line 140, please consider making the measured soil water retention curve (SWRC) data accessible. Peers are more interested in the raw data, which they can use to obtain parameters in other soil hydraulic models that they are interested**

AR: Currently some Ph.D. candidates are using the raw SWRC data to calculate the soil hydraulic parameters. We cannot upload all the raw SWRC datasets at this point of time. Additionally, as suggested by Reviewer 1, we have shared the raw SWRC data at eight representative sites in the study area, which has relatively complete profile SHP data and can represent the main land covers (two sites for each main land covers of Forestland, High coverage grassland, Middle coverage grassland and Barren land). The selected original measurements of the key SHP datasets for the main land covers have been uploaded to the public datasets (https://doi.org/10.5281/zenodo.6514191). Until now, both the spatial distribution of parameters of Van Genuchten model for the SWRC and the raw SWRC data of eight representative sites (five layers for each site) are provided. We will upload all the raw SWRC data once they finish their Ph.D. work, Thank you for your understanding.

**RC: In Line 270, the author used Kriging method in ArcGIS to interpolate the spatial SHPs, please specify the Kriging method (e.g., what kind of method, any covariates and spatial resolution) and describe the uncertainty of this method and the interpolated data.**

AR: As suggested by Reviewer 1, the Inverse Distance Weighted (IDW) method was adopted to generate the SHP products (with spatial resolution of 900 m, Figure 6). Furthermore, based on the cross-validation method, the uncertainty of the interpolated SHP products through the IDW method is calculated. The results are listed in Table S2 and Figure R1. Overall, our results show that the uncertainty of SHP decreases as SOC, $\alpha$, $\theta_r$, $K_S$, sand, clay, bulk, $\theta_s$, silt and $n$, which have the NRMSE (at first layer) of 90.5%, 75.1%, 64.1%, 46.7%, 45.6%, 23.2%, 22.5%, 20.6%, 19.2% and 14.2%, respectively. The above statement has been added to the revised manuscript.

[Figure]

Figure 6. The spatial distribution of soil texture (sand, silt, clay, %), bulk (g/cm$^3$), $log_{10}K_S$ ($log_{10}$ transformed saturated hydraulic conductivity, cm/day), the residual SM ($\theta_r$, cm$^3$/cm$^3$), saturated SM ($\theta_s$, cm$^3$/cm$^3$), $\alpha$ and $n$ in the study area.

☐ **Table S2. The uncertainty of the generated SHP datasets through the Inverse Distance Weighted method based on the in-situ observations.**

| metric | depth | clay | sand | silt | bulk | SOC | $K_S$ | $\theta_r$ | $\theta_s$ | $\alpha$ | $n$ |
|---|---|---|---|---|---|---|---|---|---|---|---|
| BIAS | 5 cm | 0.015 | -0.129 | 0.114 | -0.008 | 0.884 | 0.037 | -0.0007 | 0.0022 | -0.0002 | -0.0007 |
| | 15 cm | 0.173 | -0.742 | 0.570 | -0.029 | 0.399 | 0.001 | 0.0013 | -0.0047 | 0.0005 | 0.0088 |
| | 25 cm | -0.135 | -0.189 | 0.324 | -0.008 | 0.776 | -0.020 | -0.0056 | -0.0034 | 0.0006 | -0.0162 |
| | 40 cm | 0.200 | -0.769 | 0.569 | -0.040 | 0.414 | 0.027 | -0.0039 | 0.0007 | 0.0025 | -0.0061 |
| | 60 cm | 0.339 | -1.642 | 1.303 | -0.073 | 0.746 | 0.052 | -0.0072 | -0.0203 | 0.0014 | -0.0114 |
| PBIAS (%) | 5 cm | 0.227 | -0.444 | 0.177 | -0.642 | 23.00 | 2.625 | -0.730 | 0.420 | -0.754 | -0.048 |
| | 15 cm | 2.568 | -2.807 | 0.852 | -2.573 | 11.61 | 0.092 | 1.013 | -0.920 | 1.576 | 0.630 |
| | 25 cm | -2.074 | -0.627 | 0.511 | -0.652 | 21.45 | -1.331 | -5.299 | -0.669 | 2.021 | -1.166 |
| | 40 cm | 2.991 | -2.568 | 0.898 | -3.148 | 19.16 | 2.043 | -3.077 | 0.137 | 8.609 | -0.441 |
| | 60 cm | 5.143 | -5.960 | 1.979 | -5.474 | 32.50 | 5.303 | -5.588 | -4.001 | 6.056 | -0.845 |
| NRMSE (%) | 5 cm | 23.16 | 45.59 | 19.19 | 22.47 | 90.51 | 46.72 | 64.11 | 20.58 | 75.12 | 14.16 |
| | 15 cm | 21.56 | 53.57 | 19.73 | 20.60 | 72.74 | 44.21 | 41.00 | 26.69 | 42.29 | 11.30 |
| | 25 cm | 24.55 | 45.15 | 20.28 | 27.06 | 94.20 | 48.05 | 79.54 | 28.69 | 68.58 | 13.19 |
| | 40 cm | 22.28 | 60.95 | 27.40 | 22.58 | 114.7 | 46.30 | 64.87 | 20.96 | 39.90 | 13.15 |
| | 60 cm | 31.93 | 55.95 | 22.39 | 22.53 | 121.5 | 79.81 | 66.54 | 26.88 | 68.62 | 12.34 |

[Figure]

Figure R1. The uncertainty (NRMSE and PBIAS) of the generated SHP datasets through the Inverse Distance Weighted method based on the in-situ SHP observations.

**RC: Please explain Figure 7b.**

AR: Figure 7b represents the distribution of soil moisture range at different matrix heads for each soil layer. As the SWRC can be explained based on the parameters of Van Genuchten model. Specifically, $\theta_r$ and $\theta_s$ can reflect the lower boundary and upper boundary of the SWRC, respectively, $n$ and $\alpha$ can reflect the shape and position of the curve, respectively (Mohawesh, 2014; Assouline et al., 2021). Figure 7b can be explained by the variation of the four parameters with depth (Figure 7a). It shows that the $\theta_s$ decreases from 5 cm depth (0.55 cm$^3$/cm$^3$) to 40 cm depth (0.46 cm$^3$/cm$^3$), then increases from 40 cm to 60 cm depth (0.50 cm$^3$/cm$^3$). The $\theta_r$ fluctuation increases with depth from 5 cm depth (0.11 cm$^3$/cm$^3$) to 60 cm depth (0.15 cm$^3$/cm$^3$). While $\alpha$ increases from 5 cm depth (0.022) to 40 cm depth (0.027), then decreases to 60 cm depth (0.021). $n$ fluctuation increases from 5 cm depth (1.37) to 40 cm depth (1.41), then decreases to 60 cm depth (1.39). The explanation has been revised in the manuscript.

☐ Figure 7b represents the distribution of soil moisture range at different matrix heads for each soil layer. The variation of SWRC can be explained based on the parameters of Van Genuchten model. Specifically, $\theta_r$ and $\theta_s$ can reflect the lower boundary and upper boundary of the SWRC, respectively, $n$ and $\alpha$ can reflect the shape and position of the curve, respectively (Mohawesh, 2014; Assouline et al., 2021).

**RC: The reviewer thinks that the 'dry bulk density' is measured. Please refer to this soil property as dry bulk density in the manuscript and figures.**

AR: Thank you for your comments. Yes, the bulk density is dry bulk density, which is measured using the oven-drying method (Gwenzi et al., 2011). We have changed the bulk density to dry bulk density in the revised manuscript.

**RC: Please make the symbols of and consistently used in the manuscript but also in the legend in Figures, e.g., In Figure 6, theta_s and theta_r.**

AR: Thank you for your comments. We have revised the manuscript thoroughly and made the symbols in the manuscript, figures and tables consistent. The symbols and their definitions are listed in Table R1.

**Table R1. The symbols and their definitions in the manuscript.**

| Symbol | unit | Definition |
|---|---|---|
| SM | $cm^3/cm^3$ | soil moisture |
| clay | % | soil clay content |
| silt | % | soil silt content |
| sand | % | soil sand content |
| SOC | % | soil organic carbon |
| bulk | $g/cm^3$ | soil dry bulk density |
| $K_S$ | cm/day | soil saturated hydraulic conductivity |
| $\alpha$ | $cm^{-1}$ | parameter of Van Genuchten model that related to the suction at the air entry point [a] |
| $n$ | _ | shape parameter of Van Genuchten model [a] |
| $\theta_r$ | $cm^3/cm^3$ | residual soil moisture |
| $\theta_s$ | $cm^3/cm^3$ | saturated soil moisture |

[a] the definitions come from Assouline (2021)

**RC: L60, please delete the "." after "soil"**

AR: Thank you, we have revised it.

☐ SM can generally only be retrieved for the uppermost 5 cm of the soil (Xing et al., 2021; Zhang et al., 2019).

**RC: L82, Please specify the data, in-situ meteorologic data or other type of data?**

AR: It's from the in-situ meteorological data in the study area (He et al., 2018). We have specified the type of data in the revised manuscript.

**RC: L96, Please add one sentence to explain why you divide the area into 32?**

AR: We divided the study area into 32 homogeneous zones based on the following main factors: 1). Adequate representation of the main types of land use/land cover, soil, and topography of the study area; 2). Relatively large, homogeneous zones of the LULC, soil, and topography (DEM); and 3). Constraints of project budget and personnel in carrying out the field work. The GIS analysis procedure is as follows: 1). Convert the collected land use/land cover (LULC), soil type and digital elevation model (DEM) datasets of the study area to ArcGIS (Environmental Systems Research Institute, 2012) shapefile format; 2). Overlay the aforementioned datasets to define LULC-soil-DEM classes (polygons); 3). Aggregate those similar LULC-soil-DEM classes to produce relatively large, homogeneous classes (ESRI, 2012). After the procedure, we got 32 main LULC-soil-DEM classes (Figure 1) in the study area. The details of the procedure can be found in Jin et al. (2015). The above explanation has been added to the revised manuscript.

☐ We divided the study area into 32 homogeneous zones based on the following main factors: 1). Adequate representation of the main types of land use/land cover, soil, and topography of the study area; 2). Relatively large, homogeneous zones of the LULC, soil, and topography (DEM); and 3). Constraints of project budget and personnel in carrying out the field work. The GIS analysis procedure is as follows: 1). Convert the collected land use/land cover (LULC), soil type and digital elevation model (DEM) datasets of the study area to ArcGIS (Environmental Systems Research Institute, 2012) shapefile format; 2). Overlay the aforementioned datasets to define LULC-soil-DEM classes (polygons); 3). Aggregate those similar LULC-soil-DEM classes to produce relatively large, homogeneous classes (ESRI, 2012). After the procedure, we got 32 main LULC-soil-DEM classes (Figure 1) in the study area.

RC: **L110, Please clarify its representative on each zone.**

AR: Thank you for your comments. After getting the spatial distribution of 32 main LULC-soil-DEM zones in the study area, we selected the specific locations of representative sites for each zone based on following principles: 1). The representative site is located near the center of the largest patch for

each zone. 2). Each location is accessible within the walking distance from rural road for the installation and long-term maintenance of the equipment in the harsh high and cold mountainous areas. Despite this consideration, we had to walk 3 hours each way to get to two of our monitoring sites from rural road (Figure R2). 3). Representation of the surrounding environment. 4) Each location is some distance away from rural road to minimize the impact of human interference and keep the equipment safe. Unfortunately, we still suffered destruction and theft of our equipment in several locations, incurring several thousand dollars of loss (Figure R2). The explanation has been added to the revised manuscript.

☐ After getting the spatial distribution of 32 main LULC-soil-DEM zones in the study area, we selected the specific locations of representative sites for each zone based on following principles: 1). The representative site is located near the center of the largest patch for each zone. 2). Each location is accessible within the walking distance from rural road for the installation and long-term maintenance of the equipment in the harsh high and cold mountainous areas. 3). Representation of the surrounding environment. 4) Each location is some distance away from rural road to minimize the impact of human interference and keep the equipment safe.

[Figure]

Figure R2. (a) Datalogger damage caused by water ingress, (b) the stolen of 5TE sensor and datalogger in the field (only the broken white waterproof box remains), (c)-(d) the destruction of 5TE sensor in the field. (e)-(f) two stations that need to walk 3 hours each way from rural road to collect data and maintain the instrument.

**RC: L119, Please add one sentence to explain why you want to get reader's attention on monthly data**

AR: Thank you for your comment, we have added the explanation as follows:

☐ SM variation trend during 2014-2020 and its spatial distribution in the study area are analyzed based on the in-situ measurements. As we focused on the long-term SM variation, the monthly-average SM data were applied to get robust results.

**RC: L142, dry soil bulk density**

AR: Thank you, we have changed it to soil dry bulk density.

**RC: L145, Please clarify why 6359 hPa and the interval of matrix potential measured in your experiment? Please make measured SWRCC data accessible.**

AR: We used the refrigerated centrifuge method (CR-GIII High-Speed Refrigerated Centrifuge, Hitachi, Ltd. Figure R3) to measure the SWRC. The limitation of the maximum rotation speed is 8200 rpm, and the corresponding maximum matrix potential is 6359 hPa. The SWRC curve was measured at the matrix potentials of 9 hPa, 91 hPa, 273 hPa, 454 hPa, 726 hPa, 909 hPa, 2657 hPa, 2542 hPa, and 6359 hPa, respectively. The above description has been added to the revised manuscript. As the Ph.D. candidates are still using the raw measured SWRC data (measured soil water content at different matrix potentials), we cannot upload all the raw SWRC datasets at this point of time. However, as stated above, the raw SWRC datasets of eight representative sites for the main land covers have been uploaded to the public datasets of this manuscript (https://doi.org/10.5281/zenodo.6514191). We are willing to share all the SWRC datasets later once the Ph.D. candidates complete their theses. Thank you for your understanding.

☐ Limited by the measurement method (CR-GIII High-Speed Refrigerated Centrifuge, Hitachi, Ltd.), the maximum matrix potential of the measurement is 6359 hPa. Specifically, the SWRC curve was measured at the matrix potentials of 9 hPa, 91 hPa, 273 hPa, 454 hPa, 726 hPa, 909 hPa, 2657 hPa, 2542 hPa, and 6359 hPa, respectively.

[Figure]

Figure R3. Picture of CR-GIII High-Speed Refrigerated Centrifuge

**RC: L203, Please give the reference**

AR: Thank you for your reminder. We have added the reference in the revised manuscript: (Derrac et al., 2011)

**RC: L207, Please clarify the used interpolation method and the possible uncertainty.**

AR: As suggested by Reviewer 1, the Inverse Distance Weighted method was adopted to get the spatial distribution of SHP products. The uncertainty has been stated above.

**RC: L210-214, Please consider to move these contents in the field sampling. The reviewer is more interested to see the spatial/profile distribution of measured basis soil properties in Results.**

AR: Thank you, we have moved this paragraph to the field sampling part in the revised manuscript.

**RC: L214, Please clarify how these results are related to the information of .nc data.**

AR: The .nc data is generated based on all the in-situ measurements of SHP datasets. The specific numbers of each soil property that were used to generate the .nc data (Table S3) have been added to the revised manuscript.

☐ **Table S3. The specific number of the measured soil properties in the datasets.**

| depth/cm | clay | silt | sand | bulk | SOC | $K_S$ | SWRC |
|:---:|:---:|:---:|:---:|:---:|:---:|:---:|:---:|
| 5 | 198 | 198 | 198 | 158 | 30 | 174 | 140 |
| 15 | 25 | 25 | 25 | 25 | 25 | 25 | 25 |
| 25 | 162 | 162 | 162 | 64 | 30 | 65 | 56 |
| 40 | 23 | 23 | 23 | 30 | 28 | 30 | 22 |
| 60 | 21 | 21 | 21 | 30 | 24 | 24 | 19 |
| all | 429 | 429 | 429 | 307 | 137 | 318 | 262 |

Note: SWRC includes the parameters of Van Genuchten model ($\alpha$, $n$, $\theta_s$ and $\theta_r$) related to the soil water retention curve.

**RC: L225, Please correct to '$n$'**

AR: Thanks for the comment. We have revised it.

☐ and the soil hydraulic parameters ($\alpha$, $\theta_s$, $\theta_r$, and $n$) were calculated for the samples using this model.

**RC: L256, Please delete the "-" after "significant"**

AR: Thanks. We have revised it.

**RC: Tabe 3, Why is this value (Cv of theta_r at depth of 5 cm) so big? What is its indication if the reader tends to use your theta_r?**

AR: Table 3 describes the statistic values of SHP and Figure 7 shows more statistics information of SHP.

From the results, we can see that $\theta_r$ has small value and with a relatively large variation range, e.g., the mean and standard deviation of $\theta_r$ are 0.11 cm$^3$/cm$^3$ and 0.06 cm$^3$/cm$^3$, respectively. That's why the $\theta_r$ showed high $C_V$ value. Meanwhile, $\theta_r$ value determines the simulated minimum SM values in hydrological model (such as Hydrus-1D model, Simunek et al., 2005), the large $C_V$ value of $\theta_r$ indicates it has strong spatial heterogeneity in the study area. Thus, $\theta_r$ dataset with high spatial resolution is recommended for the hydrological models to minimize the influence of spatial heterogeneity of $\theta_r$ on hydrological modeling in the study area.

**RC: L289, decrease?**

AR: Thanks for your reminder. We have rewritten this sentence.

☐ The SOC has a maximum at 5 cm depth (where it has a median value of 4.02%), then fluctuation decreases from 15 cm depth (1.47%) to 60 cm depth (1.24%).

**RC: L290-291, Please check the consistency of your description.**

AR: Thanks for your reminder. We have checked the description and rewritten this sentence.

☐ The $\theta_s$ decreases from 5 cm depth (0.55 cm$^3$/cm$^3$) to 40 cm depth (0.46 cm$^3$/cm$^3$), then increases from 40 cm to 60 cm depth (0.50 cm$^3$/cm$^3$).

**RC: L407-408, Please rephrase this sentence. I did not see this paper establishing PTFs over the QTP.**

AR: We have rewritten this sentence.

☐ This study has explored the relationships among the basic soil properties (such as soil texture and dry bulk density) with the soil hydraulic properties (such as soil saturated hydraulic conductivity, soil water retention curve), such information can be used to establish pedotransfer function over the study area in the future.

**RC: L411, I do not think your monthly SM values can help on these purposes, unless you make SMST at the half-hourly scale available, which may become helpful a little bit.**

AR: These researches are based on the in-situ SM measurements. However, as stated above, the raw SM data at 30 min intervals can't be uploaded at this point of time. We are willing to share the SM datasets later once the Ph.D. candidates complete their theses. Thank you for your understanding.

**RC: L412, Please check the reference.**

AR: Zhang et al. (2017) and Zhang et al. (2019) evaluated the SMAP (Soil Moisture Active Passive) and SMOS (Soil Moisture and Ocean Salinity) products under different vegetation types in the study area based on the in-situ SM observations. Su et al. (2020) evaluated the simulated SM using Eagleson's Ecohydrological Model based on the in-situ SM observations in the study area. We have revised the sentence.

☐ for evaluating satellite-based SM products (Zhang et al., 2017; 2019) and validating the hydrological simulation (Su et al., 2020).

**RC: L451, I think 'new' is too much since Tibet-obs already provided very comprehensive dataset.**

AR: We have revised it.

☐ In summary, this study provides a unique and comprehensive dataset of in-situ measurements of soil physical properties and observations from long-term SM monitoring over the northeastern margin of the QTP area.

References:

Assouline, S.: What Can We Learn From the Water Retention Characteristic of a Soil Regarding Its Hydrological and Agricultural Functions? Review and Analysis of Actual Knowledge, Water Resour. Res., 57, e2021WR031026, https://doi.org/10.1029/2021WR031026, 2021.

Derrac, J., García, S., Molina, D., and Herrera, F.: A practical tutorial on the use of nonparametric statistical tests as a methodology for comparing evolutionary and swarm intelligence algorithms, Swarm and Evolutionary Computation, 1, 3-18, https://doi.org/10.1016/j.swevo.2011.02.002, 2011.

Gwenzi, W., Hinz, C., Holmes, K., Phillips, I. R., and Mullins, I. J.: Field-scale spatial variability of saturated hydraulic conductivity on a recently constructed artificial ecosystem, Geoderma, 166, 43-56, https://doi.org/10.1016/j.geoderma.2011.06.010, 2011.

Jin, X., Zhang, L. h., Gu, J., Zhao, C., Tian, J., and He, C. S.: Modeling the impacts of spatial heterogeneity in soil hydraulic properties on hydrological process in the upper reach of the Heihe River in the Qilian Mountains, Northwest China, Hydrol. Processes, 29, 3318-3327, https://doi.org/10.1002/hyp.10437, 2015.

Mohawesh, O. E.: Development of Pedotransfer Functions for Estimating Soil Retention Curves and Saturated Hydraulic Conductivity in Jordan Valley, Jordan Journal of Agricultural Sciences, 10, https://doi.org/10.12816/0029875, 2014.

Simunek, J., Van Genuchten, M. T., and Sejna, M.: The HYDRUS-1D software package for simulating the one-dimensional movement of water, heat, and multiple solutes in variably-saturated media, University of California-Riverside Research Reports, 3, 1-240, 2005.

Su, T., Zhang, B., He, X., Shao, R., Li, Y., Tian, J., Long, B., and He, C.: Rational Planning of Land Use Can Maintain Water Yield Without Damaging Ecological Stability in Upstream of Inland River: Case Study in the Hei River Basin of China, J. Geophys. Res. Atmos., 125, e2020JD032727, https://doi.org/10.1029/2020JD032727, 2020.

Zhang, L., He, C., and Zhang, M.: Multi-Scale Evaluation of the SMAP Product Using Sparse In-Situ Network over a High Mountainous Watershed, Northwest China, Remote Sens., 9, 1111, https://doi.org/10.3390/rs9111111, 2017.

Zhang, L., He, C., Zhang, M., and Zhu, Y.: Evaluation of the SMOS and SMAP soil moisture products under different vegetation types against two sparse in situ networks over arid mountainous watersheds, Northwest China, Sci. China Earth Sci., 62, 703-718, https://doi.org/10.1007/s11430-018-9308-9, 2019.

---

## Author Comment (AC4)

**RC: Reviewer Comment**,   AR: Author Response,   ☐ New Manuscript text

Dear Referee,

We would like to thank you very much for your effort in reviewing our manuscript. Please find our responses to your comments below. On the one hand, we have compared the spatial distribution of soil texture from HWSD and SoilGrid datasets with the observations to show where these data differ significantly. On the other hand, we have evaluated both the SMAP_L3 and SMAP_L4 soil moisture products to represent the performance of remote sensing soil moisture products.

Kind regards,

Dr. Prof. Chansheng He
(on behalf of all the authors)

**The comments of editor and reviewers are in black with bold text,** the author's answers are indicated in blue color, as well as old text passages. New text passages are indicated in green color.

**General Comments**

**The authors presented a valuable work by observing soil properties in Heihe Basin. The data set is well organized and presented, while its strength is also demonstrated by comparing it to other data sets. I think it should be accepted by ESSD.**

AR: We appreciate your positive comments and suggestions.

**Specific Comments**

**RC: I just have several minor comments about it, such as: the authors can compare the spatial distribution of the observed soil texture against HWSD and SoilGRiD, to show**

**where these data differ significantly.**

AR: Thanks for your suggestions. We have compared the spatial distribution of the observed soil texture (sand and clay) with that of HWSD and SoilGrid datasets (Figure R1). We also calculated the spatial distribution of BIAS (and PBIAS) of soil texture for the HWSD and SoilGrid datasets (Figure R2).

Overall, the spatial distribution of sand and clay from observation is lower than that of HWSD and SoilGrid datasets (Figures R1 and R2). Specifically, results indicate that SoilGrid dataset overestimates clay and sand in almost the entire region, while HWSD underestimates clay and sand in some areas of the study area. However, there is no obvious spatial pattern for the bias (BIAS and PBIAS) of the clay and sand estimation from both the HWSD and SoilGrid datasets.

[Figure]

Figure R1. Comparison of the spatial distribution of the soil texture (sand and clay, with unit of %) between observation (a-b) with the HWSD dataset (c-d) and SoilGrid dataset (e-f).

[Figure]

Figure R2. Spatial distribution of the BIAS (a-d) and PBIAS (e-h) for evaluating the spatial distribution of the errors of soil texture (sand and clay) of HWSD and SoilGrid datasets. The BIAS of sand and clay for HWSD (a-b) and SoilGrid (c-d); The PBIAS of sand and clay for HWSD (e-f) and SoilGrid (g-h).

**RC: From my understanding, GLDAS does not assimilate land surface information, and then its soil moisture is not improved too much. In contrast, SMAP-L4 is an assimilation product. So, if it is possible, the authors are suggested using some purely-remote sensing product instead of SMAP-L4.**

AR: Thank you for your suggestion. Yes, the SMAP_L4 product provides estimates of both surface and root zone SM products based on the assimilation of SMAP observations into the NASA Goddard Earth Observing System, Version 5 (GEOS-5) land data assimilation system (Reichle et al., 2017). The SMAP_L3 (Soil Moisture Active Passive Level-3) product provides a composite of daily estimates of global land surface conditions retrieved by the SMAP radiometer, which is a type of purely-remote sensing product (O'Neill et al., 2021). As we want to evaluate the SM product at both the surface and subsurface layers, we evaluated the SMAP_L4, ERA5-Land and GLDAS2.1_Noah SM products in this study.

Besides, in order to evaluate the performance of purely-remote sensing product, we also evaluated the SMAP_L3 SM product (SMAP Enhanced L3 Radiometer Global and Polar Grid Daily 9 km EASE-Grid Soil Moisture, Version 5, https://nsidc.org/data/SPL3SMP_E/versions/5, last accessed 11 May 2022) based on the in-situ surface SM observations (at 5 cm depth). Results are shown in Figure S8. Results show that the SMAP_L3 has a high R value (with the mean value of 0.54), but overestimates SM in the study area (with the mean Bias value of 0.055 $cm^3/cm^3$). Meanwhile, SMAP_L3 has the mean ubRMSE of 0.054 $cm^3/cm^3$, which is larger than the accuracy requirement of 0.04 $cm^3/cm^3$ (Chan et al., 2016). The overestimation of SM is different from our previous results in Tian et al. (2020), which found SMAP_L3 underestimated SM (with median Bias of -0.029 $cm^3/cm^3$ and ubRMSE of 0.037 $cm^3/cm^3$) during 2015-2016 in the study area. This may be because that SMAP_L3 Version 5, which was released in 2021, adopts the new baseline algorithm (Dual Channel Algorithm), which marks a departure from prior versions where the baseline algorithm was the Single Channel Algorithm-Vertical Polarization (O'Neill et al., 2021). Besides, our result is consistent with the results of Ahmad et al. (2022) at Maqu station, who found the overestimation of the original SMAP_L3 product, but the SMAP_L3 has good performance after the cumulative

distribution function (CDF) correction. Further analysis of the improvement of the SMAP_L3 product is not in the scope of this study. The evaluation of the SMAP_L3 product has been added in the revised manuscript.

[Figure]

Figure S8. (a) Scatterplot comparing the surface SM from SMAP_L3 with the observation at daily scale. (b) Metrics of different surface SM products (SMAP_L4, GLDAS2.1_Noah, ERA5-Land and SMAP_L3). Different letters indicate the significant difference between different surface SM products.

☐   Additionally, besides the SMAP_L4 surface and profile SM product, SMAP_L3 surface SM (SMAP Enhanced L3 Radiometer Global and Polar Grid Daily 9 km EASE-Grid Soil Moisture, Version 5, https://nsidc.org/data/SPL3SMP_E/versions/5, last accessed 11 May 2022) was also evaluated to represent the remote sensing SM product. Results (Figure S8) indicate that both the SMAP (SMAP_L3 and SMAP_L4) and ERA5-Land SM products have significantly high R-value than the GLDAS-Noah product, while both the SMAP and GLDAS-Noah SM products have significantly lower Bias than ERA5-Land product. However, SMAP_L3 SM product has a high mean ubRMSE of 0.054 $cm^3/cm^3$, which is larger than the accuracy requirement of 0.04 $cm^3/cm^3$ (Chan et al., 2016). The large Bias and ubRMSE value of SMAP_L3 SM product may be caused by the new baseline algorithm (Dual Channel Algorithm) adopted by the SMAP_L3 Version 5 product (released in 2021),

which marks a departure from prior versions where the baseline algorithm was the Single Channel Algorithm-Vertical Polarization (O'Neill et al., 2021). Further correction may be required for the SMAP_L3 Version 5 SM product in the study area (Ahmad et al., 2022).

**References:**

Ahmad, J. A., Forman, B. A., and Kumar, S. V.: Soil moisture estimation in South Asia via assimilation of SMAP retrievals, Hydrol. Earth Syst. Sci., 26, 2221-2243, https://doi.org/10.5194/hess-26-2221-2022, 2022.

Chan, S. K., Bindlish, R., O'Neill, P. E., Njoku, E., Jackson, T., Colliander, A., Chen, F., Burgin, M., Dunbar, S., and Piepmeier, J.: Assessment of the SMAP passive soil moisture product, IEEE Trans. Geosci. Remote Sens., 54, 4994-5007, https://doi.org/10.1109/TGRS.2016.2561938, 2016.

Reichle, R. H., Lannoy, G. J. M. D., Liu, Q., Ardizzone, J. V., Colliander, A., Conaty, A., Crow, W., Jackson, T. J., Jones, L. A., Kimball, J. S., Koster, R. D., Mahanama, S. P., Smith, E. B., Berg, A., Bircher, S., Bosch, D., Caldwell, T. G., Cosh, M., González-Zamora, Á., Collins, C. D. H., Jensen, K. H., Livingston, S., Lopez-Baeza, E., Martínez-Fernández, J., McNairn, H., Moghaddam, M., Pacheco, A., Pellarin, T., Prueger, J., Rowlandson, T., Seyfried, M., Starks, P., Su, Z., Thibeault, M., Velde, R. v. d., Walker, J., Wu, X., and Zeng, Y.: Assessment of the SMAP Level-4 Surface and Root-Zone Soil Moisture Product Using In Situ Measurements, J. Hydrometeorol., 18, 2621-2645, https://doi.org/10.1175/jhm-d-17-0063.1, 2017.

O'Neill, P. E., S. Chan, E. G. Njoku, T. Jackson, R. Bindlish, J. Chaubell, and A. Colliander. 2021. SMAP Enhanced L3 Radiometer Global and Polar Grid Daily 9 km EASE-Grid Soil Moisture, Version 5. [Indicate subset used]. Boulder, Colorado USA. NASA National Snow and Ice Data Center Distributed Active Archive Center. https://doi.org/10.5067/4DQ54OUIJ9DL

Tian, J., Han, Z., Bogena, H. R., Huisman, J. A., Montzka, C., Zhang, B., and He, C.: Estimation of subsurface soil moisture from surface soil moisture in cold mountainous areas, Hydrol. Earth Syst. Sci., 24, 4659-4674, https://doi.org/10.5194/hess-24-4659-2020, 2020.